# ERASING SPURIOUS CORRELATIONS WITH ACTIVATIONS

**Qiyuan He, Kai Xu, Angela Yao**
National University of Singapore
`qhe@u.nus.edu.sg, kxu@u.nus.edu.sg, ayao@comp.nus.edu.sg`

## ABSTRACT

Spurious correlations often arise when models associate features strongly correlated with, but not causally related to, the label *e.g.* an image classifier associates bodies of water with ducks. To mitigate spurious correlations, existing methods focus on learning unbiased representation or incorporating additional information about the correlations during training. This work removes spurious correlations by "**E**rasing **wi**th **A**ctivations" (EvA). EvA learns class-specific spurious indicator on each channel for the fully connected layer of pretrained networks. By erasing spurious connections during re-weighting, EvA achieves state-of-the-art performance across diverse datasets (6.2% relative gain on BAR and achieves 4.1% on Waterbirds). For biased datasets without any information about the spurious correlations, EvA can outperform previous methods (4.8% relative gain on Waterbirds) with 6 orders of magnitude less compute, highlighting its data and computational efficiency.

## 1 INTRODUCTION

Deep neural networks are susceptible to shortcuts and correlations with spurious features – features predictive for training but lacking genuine causation. For example, consider an image classifier for cats and dogs; if training images commonly depict dogs (but not cats) on grass, the model may misclassify a cat on grass as a dog because grass is learned as a spurious feature. "Right for the wrong reason" (Geirhos et al., 2020; Ross et al., 2017), spurious correlations may lead to unintended biases (Zech et al., 2018; Sagawa et al., 2019) and catastrophic failures in real-world environments (Adamson & Smith, 2018; Lesort, 2022).

Studies on spurious correlations and shortcuts (Zhu et al., 2021; Pezeshki et al., 2021; Bai et al., 2021b) have shown that deep networks prioritize simple and low-frequency features like backgrounds and textures (Xu et al., 2019). A straightforward approach adopted by Sagawa et al. (2019); Idrissi et al. (2022) tries to identify the spurious correlation with "unbiased" data, *e.g.*, dogs and cats with grass as background. These methods may be *data-wise expensive* depending on the amount of unbiased dataset or human annotation on spurious correlation, *i.e.*, spurious feature label. Other works (Arjovsky et al., 2019; Nam et al., 2020; Kim et al., 2022; Li et al., 2022; Tiwari & Shenoy, 2023) aim to learn invariant representations. These methods may be *computationally expensive*, as they perform multiple rounds of retraining.

Recent works (Ye et al., 2023; Kirichenko et al., 2022; Krueger et al., 2021) contend that neural networks, despite having spurious features, also learn "correct" core features; this is the case even at the final (linear) layer. Previous methods such as DFR (Kirichenko et al., 2022) use unbiased dataset to re-learns the last linear layer, freezing the feature extractor. Unfortunately, DFR's effectiveness decreases when the re-weighting dataset is small, and even worse when there is no additional unbiased dataset or when the additional dataset is biased (Ye et al., 2023), underscoring the need for both computational- and data-efficient alternatives.

Is it possible to identify spuriousness with minimal unbiased data while also preserving computational efficiency? In this work, we propose a method that is robust to a small unbiased dataset, as well as a variant that functions effectively without the need for any additional unbiased dataset and spurious feature label during training with only a few minutes training on a single GPU.

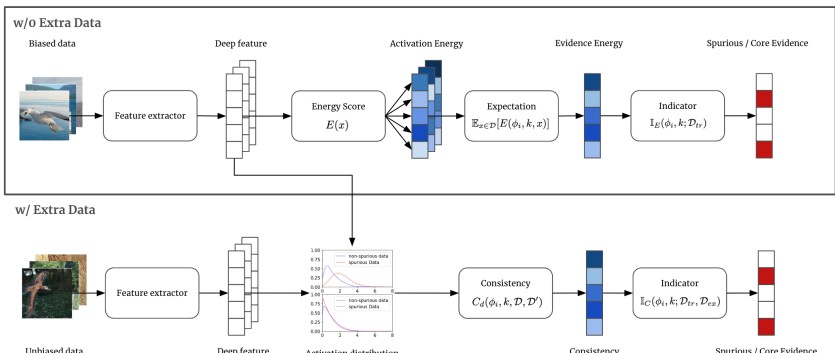

Figure 1: Spurious feature detection in EvA. When "unbiased" data is accessible, we measure the spuriousness based on the activation distribution differences between spurious and unbiased data. When only biased data is available, we estimate spuriousness by evaluating the contribution of each feature to the energy score over the biased dataset.

We focus only on spuriousness at a feature (channel) level, specifically before the last linear layer. If unbiased data is available, we compute a spurious measure for each feature based on the consistency of penultimate activations between the spurious and unbiased datasets. In the absence of unbiased data, the activations of a model fit to spurious training set can serve as a reliable indicator of spuriousness. We assess the contribution of each feature to the network's prediction confidence. Features with high confidence are likely to be spurious, as the model has over-fitted on the spurious training set; we refer to this metric as evidence energy. Subsequent to this evaluation, the fully connected layer undergoes re-weighting based on the computed spurious indicators.

Utilizing consistency and evidence energy as to quantify the extent of spuriousness, the models undergo fine-tuning based on their respective spurious indicators. "Erase Spurious Correlation with Activations" (EvA) encompasses EvA-C (EvA-consistency), tailored for scenarios with extra training data, and EvA-E (EvA-evidence energy), tailored for scenarios without extra data. Through channel-based erasure and re-weighting on the final linear layer, EvA-C necessitates only 40% of the extra unbiased data previously required and EvA-E can work even if only biased data is provided. For computational efficiency, EvA-E significantly reduces the training duration to a mere 10 minutes, in contrast to the 6 days necessitated by previous methodologies. EvA also yield improved performance outcomes across the BAR, CelebA, and Waterbirds datasets (EvA-C achieves 6.2% relative gain on BAR, EvA-E achieves 4.1% on Waterbirds, etc). Our contributions include:

- We propose a pipeline where the spurious indicators are first computed for each feature via consistency and evidence energy, the spurious correlations are then erased by re-weighting the fully connected layer based on the computed spurious indicators.

- We introduce two methods for detecting spurious feature: *Consistency* and *Evidence Energy*. The latter obviating the necessity for extra training data.

- Under both settings, training with or without extra data, EvA achieves state-of-the-art performance across various datasets with only minutes of training on a single GPU, compared to six days of training per GPU. Both versions significantly outperform previous methods without unbiased dataset or any annotation about existing spurious correlations.

## 2   RELATED WORKS

Spurious correlation has been studied from different perspectives, including worst-case generalization and deep feature re-weighting with extra data, unbiased representation learning without extra data. We provides a more detailed discussion in Appendix C.

**Worst case generalization in extra-data settings**. To address a specific spurious feature, a set of works (Sagawa et al., 2019; Bai et al., 2021a; Teney et al., 2021) explicitly divide the dataset into several feature-label groups based on the presence or absence of the given spurious feature. And the target to overcome spurious correlation is formalized as minimizing the classification error on the worst group (Sagawa et al., 2019). These methods show promising performance but are limited in practicability because of the huge cost of identifying and labeling spurious features in the real world.

**Unbiased representation learning**. To address the need for spurious feature labels, previous works try to learn unbiased representations from a biased dataset. Most of these approaches (Nam et al., 2020; Kim et al., 2022; Tiwari & Shenoy, 2023; Lee et al., 2023; Liu et al., 2021) adopt multi-stage training and introduce additional neural networks and hyperparameters, all of which require extensive computational resources for selection via validation data with less spurious correlations. Our method is more efficient by moving the debiasing module into post-hoc erasure on the linear layer.

**Deep feature reweighting**. Similar to approaches using worst case generalization, methods rooted on DFR (Kirichenko et al., 2022; Ye et al., 2023) are applied in extra-data setting where "unbiased" validation data is available during training. DFR (Kirichenko et al., 2022) is efficient by retraining the last linear layer with balanced data to robustify the classifier on overcoming spurious correlation. This supports an important assumption: deep features commonly include both spurious features and core features, motivating our works to erase spurious feature directly. However, different from our work applied in both settings, the performance of DFR decreases significantly when the extra dataset is imbalanced among feature-label groups or the size is limited.

## 3 METHOD

### 3.1 PRELIMINARIES

Following (Kirichenko et al., 2022; Ye et al., 2023), we model data-label pairs $(x, y)$ based on core and spurious components:

$$x = (x_{core}, x_{spu}) \in \mathbb{R}^d, \quad y = \beta x_{core} + \epsilon_{core}, \tag{1}$$

where core component $x_{core}$ is drawn from some distribution $\mathbb{P}$ and the spurious component $x_{spu}$ is correlated to the associated label $y$ for samples from training set $\mathcal{D}_{\text{train}}$ but not for testing set $\mathcal{D}_{\text{test}}$:

$$x_{core} \sim \mathbb{P} \in \mathbb{R}^{d_1}, \quad x_{spu} = \begin{cases} \gamma^T y + \epsilon_{spu} \in \mathbb{R}^{d_2}, \ \sim \mathcal{D}_{\text{train}} \\ \epsilon_{spu} \in \mathbb{R}^{d_2}, \ \sim \mathcal{D}_{\text{test}} \end{cases}. \tag{2}$$

In Equation 1 and 2, the terms $\epsilon_{core}$ and $\epsilon_{spu}$ denote noise associated with the core and spurious components respectively, while the parameters $\beta$ and $\gamma$ are normalized coefficients. Furthermore, the spurious component, being dependent on label $y$, is class-specific.

Consider a $k$-class neural network classifier $f : x \to \hat{y}$, where $x \in \mathbb{R}^d$, and $\hat{y} \in \mathbb{R}^k$. $f$ can be decomposed into a feature extractor $g$ and classifier $h$, *i.e.* $f = h \circ g$. The feature extractor $g : x \to Z$ yields an $m$-dimensional feature $Z \subset \mathbb{R}^m$, while the classifier $h$ is typically a linear function:

$$\hat{y} = h(Z) = WZ + b, \quad \text{where } W \in \mathbb{R}^{k \times m}, \ b \in \mathbb{R}^k. \tag{3}$$

We define $\phi$ as the distribution of $Z$ over some dataset, with $\phi^{(i)}$ representing the distribution of the $i$-th channel of $Z$. To further characterize the distribution for each class, we use $\phi^{(ik)}$ to denote the distribution $\phi^{(i)}$ for class $k$, where the input data is restricted to class $k$. If $f$ is learned on a training set contains spurious correlation for class $k$, then $\phi^{(ik)}$ will be spuriously correlated with class $k$.

### 3.2 QUANTIFYING SPURIOUS CORRELATIONS.

We aim to quantify the extent of the spurious correlation for feature $\phi^{(ik)}$. To that end, we introduce the **Consistency** measure and the **Evidence Energy** measure. The former is computed when unbiased data $\mathcal{D}_{\text{unbiased}}$ is available, while the latter can be estimated *without* unbiased data $\mathcal{D}_{\text{unbiased}}$.

#### 3.2.1 WITH UNBIASED DATA: CONSISTENCY

We define the Consistency between $\phi^{(ik)}$ and class $k$ as the distance between training and testing distributions of $\phi^{(ik)}$:

$$\mathbf{C}^{(ik)} = -\mathbf{d}(\phi_{\text{train}}^{(ik)}, \phi_{\text{test}}^{(ik)}) \approx -\mathbf{d}(\phi_{\text{train}}^{(ik)}, \phi_{\text{unbiased}}^{(ik)}). \tag{4}$$

Above, $\mathbf{d}$ is a symmetric function evaluating the distance between two distributions as a positive real number; we use the Wasserstein distance (Vallender, 1974), which is commonly used as a distribution metric (Arjovsky et al., 2017; Wang et al., 2021).

To approximate the unknown $\phi_{\text{test}}^{(ik)}$, we instead compute the distance based on $\phi_{\text{unbiased}}^{(ik)}$, estimated from additional unbiased data $\mathcal{D}_{\text{unbiased}}$. In comparison, other approaches use $\mathcal{D}_{\text{unbiased}}$ to re-learn weights (DFR (Kirichenko et al., 2022)) or select hyperparameters (Arjovsky et al., 2019; Nam et al., 2020; Kim et al., 2022; Li et al., 2022; Tiwari & Shenoy, 2023).

### 3.2.2 WITHOUT UNBIASED DATA: EVIDENCE ENERGY

We also define a measure without $\mathcal{D}_{\text{unbiased}}$ based on the observation that there is a high probability of spurious correlations for network predictions with high confidence or prediction logits (Tiwari & Shenoy, 2023; Xu et al., 2019). We elaborate on this observation from a theoretical perspective in Section 3.4 and empirically in Section 4.2. In light of this observation, our aim is to determine the feature contribution to the final logit, based on the energy.

**Energy:** The energy (LeCun et al., 2006; Liu et al., 2020) for an input $x$ is formally defined as:

$$E(x) = -T \cdot \log \sum_{k=1}^{K} e^{f_k(x)/T}, \tag{5}$$

where $T$ represents a temperature parameter, and $f_k(x)$ denotes the logit associated with class $k$. Throughout training, the energy of training samples is implicitly minimized. As such, the energy serves as an indicator of the network's prediction confidence (LeCun et al., 2006; Liu et al., 2020); the lower the energy, the more confident the prediction is.

**Evidence Energy:** We introduce the concept of evidence energy to quantify the correlation between activation $\phi^{(ik)}$ and the prediction confidence. To delineate, for a given data point $x$ from the training set, the evidence energy of $x$ is approximated as the linear term of a Taylor expansion:

$$\mathbf{e}^{(ik)}(x) \approx \frac{\delta E(x)}{\delta \phi^{(ik)}} \cdot \phi^{(ik)} = -\frac{M_k \cdot W_{ki}}{\sum_{t=1}^{K} M_t} \cdot \phi^{(ik)}, \text{ where } M_t = \exp(\sum_{j=1}^{d} W_{tj} \cdot \phi^{(jk)} + b_t). \tag{6}$$

We further define the expectation of the evidence energy on training dataset $\mathcal{D}_{\text{train}}$ as:

$$\mathbf{E}^{(ik)} = \mathbb{E}_{x \in \mathcal{D}_{\text{train}}^k}[\mathbf{e}^{(ik)}(x)] \approx \frac{1}{|\mathcal{D}_{\text{train}}^k|} \sum_{x \in \mathcal{D}_{\text{train}}^k} \mathbf{e}^{(ik)}(x), \tag{7}$$

where $\mathcal{D}_{\text{train}}^k$ denotes training data for class $k$. The evidence energy is directly correlated with consistency when the network $f$ is trained on biased data. This observation is grounded in the notion that neural networks prefers to learn simple features – the simplicity bias, as established by existing works (Xu et al., 2019; Zhu et al., 2021; Sagawa et al., 2019). Furthermore, spurious features are often simple (Pezeshki et al., 2021; Ye et al., 2023; Tiwari & Shenoy, 2023). Subsequently, during inference, spurious correlations leads to elevated confidences and therefore evidence energy scores.

### 3.3 EvA PIPELINE.

EvA is a post-hoc method that detects spurious feature, erases it by setting the corresponding weights to 0, and then re-weights the remaining core feature.

**Detection.** Given a spurious model $f$, we define a spurious feature indicator $\mathbb{I}^{(ik)}$ for feature $i$ on class $k$ to identify a set of spurious feature:

$$\Phi_k = \{\phi^j \in \Phi \mid \mathbb{I}^{(jk)} = 1\} \tag{8}$$

If $\mathcal{D}_{\text{unbiased}}$ is available, we use consistency as an indicator; otherwise, we use the evidence energy from Equation 7:

$$\mathbb{I}_{\mathbf{C}}^{(ik)} = \begin{cases} 1, & \mathbf{C}^{(ik)} \le \delta_{\mathbf{C}} \\ 0, & \mathbf{C}^{(ik)} > \delta_{\mathbf{C}} \end{cases} \quad \text{and} \quad \mathbb{I}_{\mathbf{E}}^{(ik)} = \begin{cases} 1, & \mathbf{E}^{(ik)} \le \delta_{\mathbf{E}} \\ 0, & \mathbf{E}^{(ik)} > \delta_{\mathbf{E}} \end{cases}. \tag{9}$$

Both the threshold for consistency $\delta_{\mathbf{C}}$ and evidence energy $\delta_{\mathbf{E}}$ are determined by the *erase ratio* $\epsilon$, a hyperparameter determined in the re-weighting stage.

**Reweighting.** Re-weighting limits the linear layer's dependency on spurious features and prioritizes the core features. After distinguishing the two with the above indicators, we eliminate potential spurious features within each class $k$ by setting the corresponding weights to zero before relearning the remaining non-zero weights.

More formally, given the set of spurious evidence for each class $\Phi_k$, as determined by Equation 8, we solve for new weights $\{W^*, b^*\}$, which rely only on core evidence $\Phi \setminus \Phi_k$ for each class $k$. Given a reweighting dataset $\mathcal{D}_r$, and a frozen feature extractor $g$, the objective of the linear layer $\{W^*, b^*\}$ can be defined as a constrained optimization:

$$\min_{W*, b*} \quad L(\mathcal{D}_r; g, W^*, b^*) \qquad s.t. \; W_{kj} = 0, \; \forall \phi^j \in \Phi_k, k \in \{1 \dots K\}. \tag{10}$$

$\mathcal{D}_{\text{unbiased}}$ is used as $\mathcal{D}_r$ if it is available (EvA-C); note that differs from DFR (Kirichenko et al., 2022), which retrains all the weights of the linear layer from both core and spurious features. Without any $\mathcal{D}_{\text{unbiased}}$, we use $\mathcal{D}_{\text{train}}$ instead (Eva-E). Such a setting is appealing as no extra unbiased dataset or spurious feature label is required during training. We provide the whole algorithm in Appendix B.

## 3.4 THEORETICAL RESULTS

Following the framework used by (Arjovsky et al., 2019; Ye et al., 2023), we conceptualize the network with two linear layers and employ data generation mechanism as the same as Section 3.1 for formal theoretical analysis, with further details of these conventional assumptions provided in Appendix A. In this section, we consider some class $k$ and two features $\phi^{(ak)}$ and $\phi^{(bk)}$. For simplicity, we omit the superscript $k$, and two features can be represented as:

$$\phi^{(a)} = g_{core}^{(a)} x_{core} + g_{spu}^{(a)} x_{spu}, \quad \text{and } \phi^{(b)} = g_{core}^{(b)} x_{core} + g_{spu}^{(b)} x_{spu}, \tag{11}$$

where $g_{core}^{(a)}$ is the coefficient associated with core component $x_{core}$ for feature $\phi^{(a)}$, $g_{spu}^{(a)}$ is the coefficient associated with spurious component $x_{spu}$. Similarly, $g_{core}^{(b)}$ and $g_{spu}^{(b)}$ are the corresponding coefficients of feature $\phi^{(b)}$. We analyze the relationship of consistency and evidence energy with the spurious component $x_{spu}$.

**Consistency:** Feature with low consistency can be formally interpreted as having a greater reliance on spurious components.

**Theorem 1.** *(Informal) Under a set of assumption, the consistency of $\phi^{(a)}$ exceeds that of $\phi^{(b)}$ (i.e., $\mathbf{C}^{(ak)} > \mathbf{C}^{(bk)}$) if and only if $|g_{spu}^{(a)} \mathbb{E}[x_{spu}]| < |g_{spu}^{(b)} \mathbb{E}[x_{spu}]|$.*

This theorem substantiates the claim that the spuriousness of a feature can be quantified through consistency, as higher consistency indicates reduced reliance, i.e., $|g_{spu} \mathbb{E}[x_{spu}]|$, on spurious component $x_{spu}$. The proof and more detailed explanations are given in Appendix A.1.

**Evidence Energy:** By comparing the evidence energy of two features, we demonstrate that spurious features tend to have lower evidence energy, and this tendency becomes more pronounced as the dataset becomes more biased. Let $\eta_{core}^2 I$ and $\eta_{spu}^2 I$ denote the variance of $x_{core}$ and $x_{spu}$ respectively.

**Theorem 2.** *(Informal) Under a set of assumption, if $\eta_{spu}^2 < \eta_{core}^2$ and $W_{ka} g_{spu}^{(a)} x_{spu} > W_{kb} g_{spu}^{(b)} x_{spu}$, it holds that $P\left(\mathbf{E}^{(ak)} < \mathbf{E}^{(bk)}\right) = 0.5 + R(\eta_{spu}, \eta_{core})$, where $R(\cdot, \cdot)$ is a strictly positive-valued function.*

The condition $\eta_{spu} < \eta_{core}$ mathematically describe that the noise associated with the spurious component is less than that of the core component and making the spurious feature easier to learn (Ye et al., 2023). Theorem 2 shows that if a feature contributing to the final result relies more on $x_{spu}$, i.e. $W_k g_{spu} x_{spu}$, it's more likely to have lower evidence energy. The residual $R$ increases when the noise in the spurious feature is significantly smaller than the noise in the core feature, *i.e.*, the spurious feature is easier for the model to learn. The proof and more details for Theorem 2 are given in Appendix A.2.

While the theoretical model is a simplified view of neural networks and real-world data, we verify empirically in Section 4.2 that both consistency and evidence energy can identify spurious features.

# 4 EXPERIMENTS & ANALYSIS

## 4.1 EXPERIMENTAL SETUP

**Datasets and Evaluation**. We verify our proposed framework with the datasets listed in Table 1. The datasets can be characterized by the *conflicting ratio* (Tiwari & Shenoy, 2023), or the proportion of samples that counter the spurious correlation within the entire dataset. A detailed description is given in Appendix D. For each dataset, we report the mean and standard deviation over ten runs on the average top-1 **Accuracy** unless otherwise indicated.

- **CMNIST**. Color-MNIST (Tiwari & Shenoy, 2023) is a synthesized dataset that colors the digits 0 and 1 from MNIST (Deng, 2012). In the training set, the 0's are red and the 1's are green; in testing, the colors are mixed across the two digits.

- **BAR**. The Biased Activity Recognition dataset (Nam et al., 2020), features six human actions spuriously correlated with the background in training but not in testing.

- **Waterbirds**. Waterbirds (Sagawa et al., 2019) is a two-class image dataset of water- and ground-birds; the class is correlated with the background (water or ground) in training but not in testing. It is evaluated according to **Accuracy** and **Worst Acc** (Sagawa et al., 2019; Ye et al., 2021; 2023), which is the accuracy of the worst label-context group.

- **CelebA Hair**. CelebA is a large human face benchmark (Liu et al., 2018). Following settings from (Ye et al., 2023; Tiwari & Shenoy, 2023), the task is to predict hair color based on the face image. In the training dataset, >99% instances with blond hair are women so blond hair and gender are spuriously correlated. CelebA is evaluated additionally based on the **Unbiased** accuracy over each label-context group and **Conflicting** accuracy, averaged over the bias-conflicting samples per class following (Li et al., 2022; Kim et al., 2022).

**Training and Validation**. For fair comparison with previous works (Nam et al., 2020; Tiwari & Shenoy, 2023; Li et al., 2022; Teney et al., 2022), we use the ResNet18 (He et al., 2016) as the base model initialized with weights pre-trained on ImageNet and the same settings of (Nam et al., 2020; Tiwari & Shenoy, 2023; Li et al., 2022; Teney et al., 2022). To compare with the methods focusing on minimizing the error in the worst label-context group, we also provide the results on ResNet50 with previous work in the Appendix E.2.

We use SGD optimization with a fixed learning rate of 0.001. For CMNIST, Waterbirds and CelebA, we use the same validation and test dataset as (Sagawa et al., 2019; Ye et al., 2023; Tiwari & Shenoy, 2023). BAR has no provided validation set, so we randomly split the testing dataset into two equal halves across the ten experiments to form a validation set that follows the same testing distribution. To select the erase ratio $\epsilon$, we retrain the linear layer with different erase ratio candidates and select the one with the highest accuracy on $\mathcal{D}_{\text{unbiased}}$.

**Reweighting dataset**. Under the setting without extra data, the training dataset is used as a reweighting dataset. Under the setting with extra data, the validation dataset is split equally, while one serves as a reweighting dataset, the other serves to select hyper-parameter. In BAR where the spurious feature is not labeled, the reweighting dataset is also biased. In CelebA and Waterbirds, we follow DFR (Kirichenko et al., 2022) to sample "unbiased" dataset from the validation set.

## 4.2 ANALYSIS

We analyze our methods according to 1) the correlation between evidence energy and consistency, 2) the effectiveness of feature erasure, 3) data efficiency, and 4) computation efficiency.

**Evidence energy & Consistency.** We empirically access the correlation between evidence energy and consistency. In Figure 2, we present an illustrative example of a feature set predicting the action class `Throwing` in the BAR dataset, with an erase ratio of $\epsilon = 0.1$. Inconsistency is computed between the training dataset and the validation dataset. Figure 2 (a) shows a moderate positive correlation

| Dataset | Val ratio | Hparams goal | Conflicting ratio | |
|---|---|---|---|---|
| | | | Train | Val / Test |
| CMNIST | 30% | **Accuracy** | 0% | 50% |
| BAR | 16% | **Accuracy** | 0% | 100% |
| Waterbirds | 25% | **Worst Acc** | 5% | 25% |
| CelebA | 12% | **Unbiased** | 0.8% | 0.9% |

Table 1: Summary of dataset details. 'Val ratio' indicates the proportion of the validation dataset size relative to the training dataset size. 'Conflicting ratio' denotes the proportion of images that counter the spurious correlation within the entire dataset.

| Method | Target Feature | Color Accuracy | Digit Accuracy |
|---|---|---|---|
| EvA | Color | $\mathbf{98.74}_{\pm 1.0}$ | $52.34_{\pm 8.8}$ |
| | Digit | $51.45_{\pm 2.4}$ | $\mathbf{97.32}_{\pm 2.8}$ |
| ERM | Color | $97.78_{\pm 1.3}$ | $51.66_{\pm 1.8}$ |
| | Digit | $92.11_{\pm 2.5}$ | $56.03_{\pm 7.2}$ |

Table 2: Accuracy (%; mean$_{\pm \text{std}}$) on CMNIST Color Dataset and Digit Dataset. ERM fails to learn the more complex digit feature, whereas EvA achieves 97.78% on the digit dataset and only 51.45% on the color dataset, indicating erasure of the color feature.

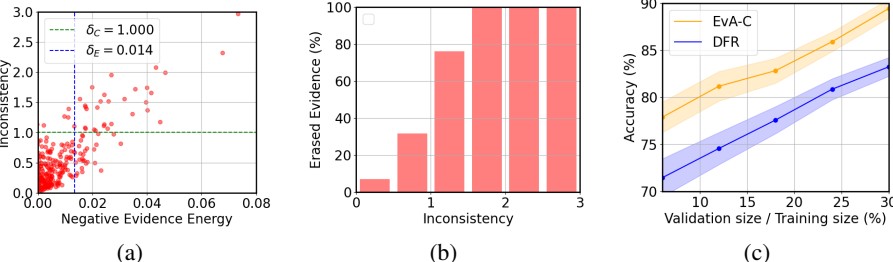

(a)       (b)       (c)

Figure 2: (a) Correlation between negative consistency and negative evidence energy for class Throwing in the BAR dataset. Each point represents feature $\phi_i$ for a sample. Feature on the right of the blue line is erased by evidence energy and most of spurious feature, *i.e.* above the green line, is erased as well. (b) Percentage of erased feature within a range of inconsistency for class Throwing in the BAR dataset, *e.g.* 78% feature with inconsistency between $[1.0, 1.5]$ is erased. Erasing by evidence energy results in erasing most feature with high inconsistency while keeping the core feature. (c) Effect of unbiased dataset size on BAR's **Accuracy**. As the unbiased set diminishes in size, the accuracy of EvA-C surpasses that of DFR by a greater margin.

between negative evidence energy and inconsistency, with a Pearson correlation coefficient of 0.704. When EvA-E removes the feature above the erase threshold, we simultaneously eliminate most spurious feature identified by consistency, which is demonstrated in Figure 2 (b). This observation underscores the effectiveness of EvA in erasing a substantial amount of spurious feature while still maintaining core feature.

We further use Integrated Gradient (Sundararajan et al., 2017) to compute the saliency map of the value of feature with respect to the input, which shows that feature with lower evidence energy and consistency tends to focus on spurious feature. We provide more analysis in the Appendix E.3.

**Effectiveness on Feature Erasure.** We verify the ability to control the feature of our EvA framework on the CMNIST Dataset, as proposed by (Tiwari & Shenoy, 2023). In the testing dataset where the colors are mixed across the two digits, we report the accuracy of predicting color labels and digit labels. If the model erases the target feature successfully, then it will have nearly 50% accuracy in predicting it but high accuracy when predicting the other feature. Notably, the target feature is determined by the validation dataset. For instance, if the target feature is color, then the label of validation data is also dependent on the color, and vice-versa.

Table 2 shows that EvA easily controls the feature it relies on to make predictions. Due to the simplicity bias (Zhu et al., 2021), empirical risk minimization (ERM) still tends to learn color features when the target is a digit, failing to overcome the spurious correlation between color and digit. EvA overcomes it by erasing the other feature, showing consistently high accuracy on the target feature.

**Data Efficiency: Size of Unbiased Dataset**. We evaluate the impact of the unbiased dataset size on test accuracy and compare with direct training on the CMNIST dataset. The size of the biased training set is fixed at 2048, while varying amounts of unbiased data are supplied to assess the minimal data required for effective debiasing. As shown in Table 3, both EvA-E and EvA-C achieve high performance even with a very small fraction of unbiased dataset (0.7% of the training set). Additional results on the Waterbirds dataset, compared with DFR, are presented in Figure 2 (c).

**Data Efficiency: Applicability to Biased Datasets without Spurious Feature Labels**. We evaluate EvA-E against two state-of-the-art approaches that do not use unbiased datasets for tuning, JTT (Liu et al., 2021) and DivDis (Lee et al., 2023), on Waterbirds using ResNet50. Following DivDis, when the validation set is unbiased, the model is tuned using the Worst Accuracy across groups. Otherwise,

| Size of unbiased dataset | 16 | 32 | 64 | 128 | 256 | 512 | 1024 |
|---|---|---|---|---|---|---|---|
| Direct Training | $57.1_{\pm1.8}$ | $59.3_{\pm9.6}$ | $61.4_{\pm9.0}$ | $62.5_{\pm5.1}$ | $78.7_{\pm2.3}$ | $88.3_{\pm0.4}$ | $97.2_{\pm1.3}$ |
| EvA-E | $92.9_{\pm0.7}$ | $93.0_{\pm0.2}$ | $93.1_{\pm0.1}$ | $93.1_{\pm0.0}$ | $93.1_{\pm0.0}$ | $93.1_{\pm0.0}$ | $93.1_{\pm0.0}$ |
| EvA-C | $91.6_{\pm2.0}$ | $96.0_{\pm0.5}$ | $97.3_{\pm0.2}$ | $97.9_{\pm0.0}$ | $98.5_{\pm0.0}$ | $98.8_{\pm0.0}$ | $98.9_{\pm0.0}$ |

Table 3: Accuracy of different methods across various sizes of provided unbiased datasets with standard deviations on CMNIST. It's noteworthy that EvA-E and EvA-C can debiased the model well even only using unbiased dataset with very small size (16).

| | JTT (Liu et al., 2021) | | DivDis (Lee et al., 2023) | | EvA-E (Ours) | |
|---|---|---|---|---|---|---|
| Tuning Data: | Unbiased | Biased | Unbiased | Biased | Unbiased | Biased |
| Worst Acc | **86.7%** | 62.5% | 85.6% | 81.0% | 86.6% | **85.8%** |

Table 4: Worst Accuracy of methods tuned on ResNet-50 with different dataset. Compared to previous state-of-the-art methods, our method EvA-E can achieve much better performance even with biased dataset (85.8% against 81.0% and 62.5%).

| Method | Extra Data | BAR Top-1 Accuracy | CelebA Unbiased | CelebA Conflicting | Waterbirds Accuracy | Waterbirds Worst Acc | FLOP |
|---|---|---|---|---|---|---|---|
| ERM | ✗ | $60.51_{\pm4.3}$ | $70.25_{\pm0.4}$ | $52.52_{\pm0.2}$ | $94.10_{\pm4.3}$ | $63.74_{\pm3.1}$ | $\geq 10^9$ |
| LfF (Nam et al., 2020) | ✗ | $62.98_{\pm2.8}$ | $84.24_{\pm0.4}$ | $81.24_{\pm1.4}$ | - | - | $\geq 10^9$ |
| LWBC (Kim et al., 2022) | ✗ | $68.45_{\pm1.3}$ | $88.90_{\pm1.6}$ | $87.22_{\pm1.1}$ | - | - | $\geq 10^9$ |
| Debian (Li et al., 2022) | ✗ | $69.88_{\pm2.9}$ | $86.74_{\pm3.2}$ | $85.33_{\pm3.7}$ | - | - | $\geq 10^9$ |
| SiFER (Tiwari & Shenoy, 2023) | ✗ | $72.08_{\pm0.4}$ | $89.00_{\pm0.9}$ | $88.04_{\pm1.3}$ | $96.11_{\pm0.6}$ | $77.22_{\pm0.4}$ | $\geq 10^9$ |
| EvA-E | ✗ | $\mathbf{73.70}_{\pm0.8}$ | $\mathbf{90.51}_{\pm1.0}$ | $\mathbf{88.74}_{\pm1.4}$ | $\mathbf{96.95}_{\pm0.9}$ | $\mathbf{81.31}_{\pm1.5}$ | $\approx 10^3$ |
| DFR (Kirichenko et al., 2022) | ✓ | $83.23_{\pm1.6}$ | $90.89_{\pm0.4}$ | $90.11_{\pm1.2}$ | $91.70_{\pm0.7}$ | $83.32_{\pm0.4}$ | $\approx 10^3$ |
| EvA-C | ✓ | $\mathbf{89.43}_{\pm1.0}$ | $\mathbf{91.32}_{\pm0.2}$ | $90.39_{\pm0.8}$ | $\mathbf{92.48}_{\pm0.1}$ | $\mathbf{86.70}_{\pm0.3}$ | $\approx 10^3$ |

Table 5: Performance (%; mean$_{\pm\text{std}}$) on CelebA / Waterbirds / BAR Test Dataset. The second column indicates if the method uses the information of validation dataset (✓) for training. Our method outperforms other methods under both settings while significantly reducing the computation cost (from $10^9$ to $10^3$). Here, we use ResNet18 for all methods for fair comparison. Note these may differ from the original paper, which report ResNet50 results; we provide ResNet-50 results on CelebA and Waterbirds in Appendix E.2.

it is tuned using the mean accuracy since spurious feature labels are unavailable in biased datasets to align with real world application. As shown in Table 4, our method performs competitively on the unbiased dataset and significantly outperforms DivDis (by 4.8%) and JTT (by 23.3%) on the biased dataset. This demonstrates that EvA-E is more data-efficient. Similarly, EvA-C is also more data-efficient than DFR, which we further show it in Appendix E.2 that EvA-C is more robust to biased dataset without spurious feature label.

**Computation Efficiency.** Most methods not relying on specific spurious feature information (Tiwari & Shenoy, 2023; Nam et al., 2020; Kim et al., 2022) need to introduce hyperparameters to control the feature. The computation expense of hyperparameter selection is commonly ignored in the previous work. However, this is serious to be considered to apply these methods in the real world. Since the selection module in EvA is based on post-hoc retraining linear layer (FLOP $\approx 10^3$), we are six orders of magnitude more efficient than other methods that require retraining the whole neural network (FLOP $\geq 10^9$) (Yu et al., 2020; Nam et al., 2020; Tiwari & Shenoy, 2023) as shown in Table 5. In our experiments, aimed at selecting the best hyperparameters from 90 candidates, EvA-E significantly reduces computation time while achieving higher accuracy within 10 minutes, compared to the 6 days required by SiFER (Tiwari & Shenoy, 2023) on a single RTX 3080 GPU. Additional discussions and detailed FLOP computations are provided in Appendix E.2.

## 4.3 Comparison to State-of-the-Art

We conducted a comprehensive comparison of our proposed methods, EvA-E and EvA-C, against state-of-the-art approaches in both with extra data and without extra data settings, as summarized in Table 5. Our methods have substantial gain whether or not extra information and data are provided. Remarkably, on the BAR dataset, which presents unique challenges due to its biased validation dataset, EvA-C outperforms DFR by a substantial margin of 6.2%. We further show the results on ResNet50 in the Appendix E.2. As we show in Section 4.2 and Appendix E.2, our methods surpass state-of-the-art methods by 4.8% (EvA-E) and 4.6% (EvA-C) on Waterbirds when no spurious feature label provided with much smaller computation cost for tuning, indicating strong improvement on computation and data efficiency.

## 4.4 Ablation Studies

**Class-wise feature detection**. EvA includes a strong consideration that each class has different spurious feature. This is important to guarantee the performance to overcome spurious feature since spurious correlation is dependent on the class. As shown in Figure 3 (b), to each pair of classes, only a small portion of erased feature is shared. Concurrently, Figure 3 (a) shows the accuracy of EvA-E

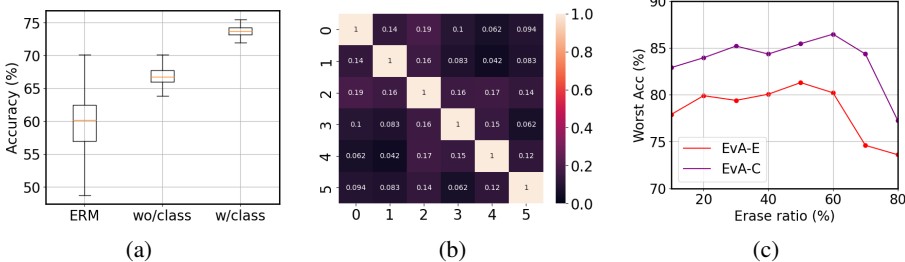

Figure 3: (a) Accuracy of different settings on the BAR dataset. The accuracy of EvA with class-wise erasure is stably higher than the one without class-wise erasure. (b) Overlapping ratio of erased feature. Illustrates the overlapping ratio of erased feature for each class. Notably, the number of erased feature instances shared between the two classes is relatively low. (c) Impact of the erase ratio on Waterbirds' **Worst Acc**. With the increment in the erase ratio ($\epsilon$), the accuracy of the worst group for both EvA-E and EvA-C exhibits a trend of gradual improvement followed by a steep decline.

on the BAR dataset, where class-wise erasure (w/class-wise) is more powerful than the one without such consideration (wo/class-wise).

**Impact of erase ratio $\epsilon$.** Figure 3 (c) illustrates the effect of the hyperparameter erase ratio $\epsilon$ on the accuracy of our model. For both EvA-E and EvA-C methods, an initial increase in the erase ratio results in a gradual improvement in the accuracy for the worst group in the Waterbirds dataset. However, this trend reverses into a sharp decline once the erase ratio exceeds a certain point—60% for EvA-C and 50% for EvA-E. This suggests that beyond these thresholds, the core feature is likely being removed, thus hindering the model's accuracy.

**Impact of unbiased dataset size**. Both EvA-C and DFR reweight based on additional data from the unbiased dataset. As Figure 2 (c) shows, the accuracy of EvA-C and DFR both increase when the unbiased dataset gets larger, though EvA-C consistently outperforms DFR. EvA-C requires less data than DFR to have a relatively good improvement. We provide more ablation studies in Appendix. E.4.

## 5 CONCLUSION

To address spurious correlation, we focus on each channel of deep feature, and formally define spurious indicators including consistency if the unbiased data is available and evidence energy when there is no unbiased dataset. Our findings demonstrate that erasing potential spurious feature and reweighting on core feature can mitigate spurious correlations, irrespective of the availability of additional information.

The proposed method, Erase spurious correlation with Activation (EvA), is distinguished by its remarkable efficiency and effectiveness in addressing spurious correlations in practical scenarios.

- **Data efficiency**. A primary advantage of EvA is its superior data efficiency. EvA-C outperforms deep feature reweighting (DFR) methodologies, yielding better results with fewer unbiased dataset or even biased dataset. When no unbiased dataset and spurious feature label is provided, EvA-E offers an alternative to debias deep learning models. Overall, EvA demands less data to achieve competitive outcomes in mitigating spurious correlations.

- **Computational efficiency**. Our method boasts computational efficiency, requiring only one hyperparameter, the erase ratio, during the erasing phase. Unlike conventional debiasing techniques that typically depend on complex network architectures, EvA simplifies the process by relying on alternative linear probing.

In summary, the EvA methodology heralds a potent solution for combating spurious correlations, with its efficiency and adaptability to real world. By utilizing indicators including consistency and evidence energy, it identifies and erases spurious feature, thereby elevating the bias. Our contributions pave the way for future investigation into activation-based spurious feature detection and erasure, offering a fresh perspective to overcome the spurious correlation problem.

**Acknowledgments** This research was funded by the NUS Artificial Intelligence Institute (NAII) seed grant number NAII-SF-2024-003.

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

In the supplementary material, we present

- Section A: Theoretical analysis of evidence energy and consistency in Section 3.4.
- Section B: Algorithm of EvA with pseudo code.
- Section C: More discussion on the related work and the main difference with our methods.
- Section D: More detailed description of dataset utilized in the experiments.
- Section E: Supplementary experiments results with 1) More details on the results of BAR dataset; 2) Experiments on ResNet50 mentioned in Section 4.1 further demonstrating computation efficiency and data efficiency of our methods; 3) Additional analysis on detected spurious feature including activation distribution and saliency map; 4) Additional ablation study with Lasso Regression and other possible indicators.

Note that all the notations and abbreviations here are consistent with the main manuscript.

## A  THEORETICAL ANALYSIS

In this section, we conduct a formal analysis of the characteristics pertaining to consistency and evidence energy, under reasonable assumptions. Our approach aligns with the frameworks established in Ye et al. (2023); Arjovsky et al. (2019). In this setting, the data tuple $(x, y)$ is generated via the following mechanism:

$$
\begin{aligned}
x_{core} &\sim \mathbb{P} \in \mathbb{R}^{d_1 \times 1}, \\
y &= \beta x_{core} + \epsilon_{core}, \\
x_{spu} &= \begin{cases} \gamma^T y + \epsilon_{spu} \in \mathbb{R}^{d_2 \times 1}, \sim \mathcal{D}_{\text{train}} \\ \epsilon_{spu} \in \mathbb{R}^{d_2 \times 1}, \sim \mathcal{D}_{\text{test}} \end{cases} \\
x &= (x_{core}, x_{spu}) \in \mathbb{R}^{d \times 1}
\end{aligned}
\tag{12}
$$

Here, $x_{core}$ represents the core component, while $x_{spu}$ denotes the spurious component, which exhibits different distributions in $\mathcal{D}_{\text{train}}$ and $\mathcal{D}_{\text{test}}$. The term $\epsilon_{core}$ signifies the noise associated with the core component, possessing a variance of $\eta_{core}^2$, and $\epsilon_{spu}$ represents the noise linked to the spurious component, with a variance of $\eta_{spu}^2$. Both noise terms have a mean of zero. The parameters $\beta$ and $\gamma$ are normalized coefficients, each with a unit $l_2$ norm.

In our study, we conceptualize the network using two key components: a feature extractor $g$ and a linear probing mechanism $h$. Simplifying our approach, we treat $g$ as a linear layer. The activation function $\phi^{(i)}$ is then defined as a linear transformation $g$ applied to the input vector $x = (x_{core}, x_{spu})$. This simplification is aligned with prevalent theoretical models employed in the analysis of non-convex optimization challenges in deep learning, as elaborated in references Arora et al. (2018); Kumar et al. (2022); Ye et al. (2023). Additionally, we base our analysis on the assumption that the training ($\mathcal{D}_{\text{train}}$) and testing ($\mathcal{D}_{\text{test}}$) datasets are infinitely large and that training is conducted with an infinitesimally small learning rate. This perspective enables us to distinctly represent the model in terms of core and spurious features.

$$
\phi^{(i)}(x) = g_{core}^{(i)} x_{core} + g_{spu}^{(i)} x_{spu},
\tag{13}
$$

where $i$ denotes the feature index, with a total count of $m$ feature. Derived from the training dataset $\mathcal{D}_{\text{train}}$, the overall model can be expressed as:

$$
\hat{y} = v_{core} x_{core} + v_{spu} x_{spu}
\tag{14}
$$

$$
v_{core} = \sum_{i=1}^{m} W_{ki} \cdot g_{core}^{(i)},
\tag{15}
$$

$$
v_{spu} = \sum_{i=1}^{m} W_{ki} \cdot g_{spu}^{(i)}
\tag{16}
$$

### A.1 ANALYSIS ON CONSISTENCY

In our analysis, we demonstrate that feature with low consistency can be formally interpreted as having a greater reliance on spurious features, assuming the random variables follow a Gaussian distribution. To substantiate this claim, we reference established conclusions regarding the Wasserstein distance between two Gaussian distributions as detailed in Takatsu (2011); Salmona et al. (2021).

**Lemma 1:** Consider two independent random variables $x_1 \sim N(\mu_1, \Sigma_1)$ and $x_2 \sim N(\mu_2, \Sigma_2)$, each following a Gaussian distribution. The 2-Wasserstein distance between them is given by:

$$d^2 = |\mu_1 - \mu_2|^2 + \text{Tr}(\Sigma_1 + \Sigma_2 - 2(\Sigma_1^{1/2}\Sigma_2\Sigma_1^{1/2})^{1/2}) \tag{17}$$

Building upon this lemma, we present the following theorem:

**Theorem 1:** Given a data distribution:

$$((x_{core}, x_{spu}), y) \sim \mathcal{D}_{\text{train}},$$
$$((x_{core}, x'_{spu}), y) \sim \mathcal{D}_{\text{test}},$$
$$x_{core} \sim N(\mu_{core}, \Sigma^2_{core}),$$
$$\epsilon_{core} \sim N(0, \eta^2_{core}),$$
$$\epsilon_{spu} \sim N(0, \eta^2_{spu}),$$

and considering two pieces of feature $\phi^a$ and $\phi^b$:

$$\phi^{(a)}(x) = g^{(a)}_{core}x_{core} + g^{(a)}_{spu}x_{spu},$$
$$\phi^{(b)}(x) = g^{(b)}_{core}x_{core} + g^{(b)}_{spu}x_{spu},$$

The consistency of $\phi^{(a)}$ exceeds that of $\phi^{(b)}$ (i.e., $\mathbf{C}^{(ak)} > \mathbf{C}^{(bk)}$) if and only if $|g^{(a)}_{spu}\mathbf{E}[x_{spu}]| < |g^{(b)}_{spu}\mathbf{E}[x_{spu}]|$.

**Proof:** Considering the assumption of infinite data points within the datasets $\mathcal{D}_{\text{train}}$ and $\mathcal{D}_{\text{test}}$, these represent continuous joint distributions of $(x, y)$. Given that $x_{spu} = \gamma^T y + \epsilon_{spu}$ with $\epsilon_{spu} \sim N(0, \eta^2_{spu})$, we deduce:

$$x_{spu} \sim N(\gamma^T y, \eta^2_{spu}),$$
$$x'_{spu} \sim N(0, \eta^2_{spu}).$$

Since $x_{core}$ is independent of $x_{spu}$, we can further deduce:

$$\phi^{(a)}(x \mid \mathcal{D}_{\text{train}}) \sim N(g^{(a)}_{core}\mu_{core} + g^{(a)}_{spu}\gamma^T y,$$
$$g^{(a)}_{core}{}^2\Sigma^2_{core} + g^{(a)}_{spu}{}^2\eta^2_{spu}),$$
$$\phi^{(a)}(x \mid \mathcal{D}_{\text{test}}) \sim N(g^{(a)}_{core}\mu_{core},$$
$$g^{(a)}_{core}{}^2\Sigma^2_{core} + g^{(a)}_{spu}{}^2\eta^2_{spu}).$$

Combining this with Lemma 1, we establish that the consistency of feature is:

$$\mathbf{C}^{(ak)} = -|g^{(a)}_{spu}\gamma^T y|,$$
$$\mathbf{C}^{(bk)} = -|g^{(b)}_{spu}\gamma^T y|.$$

Notice that $\gamma^T y = \mathbf{E}[x_{spu}]$, it's obvious that $\mathbf{C}^{(ak)} = -|g^{(a)}_{spu}\mathbf{E}[x_{spu}]|$ and $\mathbf{C}^{(bk)} = -|g^{(b)}_{spu}\mathbf{E}[x_{spu}]|$. Therefore, it follows that $\mathbf{C}^{(ak)} > \mathbf{C}^{(bk)}$ if and only if $|g^{(a)}_{spu}\mathbf{E}[x_{spu}]| < |g^{(b)}_{spu}\mathbf{E}[x_{spu}]|$.

This theorem with an assumed model substantiates the claim that the spuriousness of feature can be quantified through consistency, as higher consistency indicates a reduced reliance on spurious features and provides insights on the more complicated scenario.

## A.2 Analysis on evidence energy

In this section, we delve into the analysis of evidence energy as a metric for estimating the spuriousness of specific feature, comparing the activation energies of two pieces of feature. Initially, let's consider the energy score of a sample derived from the Negative Log-Likelihood (NLL) Loss during training. The NLL loss, which is minimized in this process, is defined as follows:

$$\mathcal{L}\text{nll} = \mathbb{E}(x, y) \sim P^{\text{tr}} \left( -\log \frac{e^{f_y(x)/T}}{\sum_{j=1}^{K} e^{f_j(x)/T}} \right) \tag{18}$$

Given that the energy function is defined as $E(x, y) = -f_y(x)$, we can rewrite the NLL loss as:

$$\mathcal{L}\text{nll} = \mathbb{E}(x, y) \sim P^{\text{tr}} \left( \frac{1}{T} \cdot E(x, y) + \log \sum_{j=1}^{K} e^{-E(x,j)/T} \right). \tag{19}$$

The concept of evidence energy for $\phi^{(i)}(x)$ is derived from the activation's contribution to the negative free energy, expressed as:

$$E(x) = -\log \sum_{k=1}^{K} e^{-E(x,k)/T}, \tag{20}$$

and this can be approximated using a Taylor Expansion:

$$\mathbf{E}^{(ik)} \approx -\frac{e^{-E(x,k)/T} \cdot W_{ki}}{\sum_{c=1}^{C} e^{-E(x,c)/T}} \cdot \phi^{(i)}(x) \tag{21}$$

To elucidate what activation energy reveals, let us consider two pieces of feature, represented as follows:

$$\phi^{(a)}(x) = g_{core}^{(a)} x_{core} + g_{spu}^{(a)} x_{spu}, \tag{22}$$

$$\phi^{(b)}(x) = g_{core}^{(b)} x_{core} + g_{spu}^{(b)} x_{spu}. \tag{23}$$

The comparison of their relative energies can be quantified by:

$$\mathbf{E}^{(ak)} - \mathbf{E}^{(bk)} = -\frac{e^{-E(x,k)/T}}{\sum_{c=1}^{C} e^{-E(x,c)/T}} (W_{ka} \cdot \phi^{(a)}(x) - W_{kb} \cdot \phi^{(b)}(x)) \tag{24}$$

$$= -\frac{e^{-E(x,k)/T}}{\sum_{c=1}^{C} e^{-E(x,c)/T}} (W_{ka} \cdot (g_{core}^{(a)} x_{core} + g_{spu}^{(a)} x_{spu}) - W_{kb} \cdot (g_{core}^{(b)} x_{core} + g_{spu}^{(b)} x_{spu})) \tag{25}$$

$$= -\frac{e^{-E(x,k)/T}}{\sum_{c=1}^{C} e^{-E(x,c)/T}} (W_{ka} g_{core}^{(a)} x_{core} + W_{ka} g_{spu}^{(a)} x_{spu} - W_{kb} g_{core}^{(b)} x_{core} + W_{kb} g_{spu}^{(b)} x_{spu}) \tag{26}$$

This formulation allows us to quantitatively assess how different feature leverage core versus spurious features in their activation energy. Specifically, our objective is to prove the following theorem.

**Theorem 2:** Under the same condition given in **Theorem 1**, if $\eta_{spu} < \eta_{core}$ and $W_{ka} g_{spu}^{(a)} x_{spu} > W_{kb} g_{spu}^{(b)} x_{spu}$, it holds that $P\left(\mathbf{E}^{(ak)} < \mathbf{E}^{(bk)}\right) = 0.5 + R(\eta_{spu}, \eta_{core})$, where $R(\eta_{spu}, \eta_{core}) > 0$.

This theorem is particularly relevant in scenarios where the noise associated with the spurious feature is less than that of the core feature, i.e., the training dataset is biased. This indicates that spurious correlations are more easily learned. To establish the foundation for proving Theorem 2, we first introduce Lemma 2.

**Lemma 2:** Consider a model $\hat{y}^* = v_{core}^* x_{core} + v_{spu}^* x_{spu}$ trained with empirical risk minimization, perfectly matching an infinite number of data points in $\mathcal{D}_{\text{train}}$. If $\eta_{spu} < \eta_{core}$, then both the expectation and variance of the spurious component's contribution exceed those of the core component. Formally:

$$\mathbb{E}_{\mathcal{D}_{\text{train}}}^2 [v_{spu}^* x_{spu}] > \mathbb{E}_{\mathcal{D}_{\text{train}}}^2 [v_{core}^* x_{core}], \tag{27}$$

$$\text{Var}_{\mathcal{D}_{\text{train}}} [v_{spu}^* x_{spu}] > \text{Var}_{\mathcal{D}_{\text{train}}} [v_{core}^* x_{core}] \tag{28}$$

**Proof of Lemma 2:** Utilizing the conclusion from (Ye et al., 2023), we have:

$$(v_{core}^*, v_{spu}^*) = \left( \frac{\eta_{spu}^2}{\eta_{spu}^2 + \eta_{core}^2} \beta, \frac{\eta_{core}^2}{\eta_{spu}^2 + \eta_{core}^2} \gamma \right) \tag{29}$$

Considering $y = \beta x_{core} + \epsilon_{core}$ and $x_{spu} = y\gamma^T + \epsilon_{spu}$ in $D_{tr}$, we derive:

$$v_{spu}^* x_{spu} = \frac{\eta_{core}^2}{\eta_{spu}^2 + \eta_{core}^2} (y + \gamma \epsilon_{spu}),$$

$$v_{core}^* x_{core} = \frac{\eta_{spu}^2}{\eta_{spu}^2 + \eta_{core}^2} (y - \beta \epsilon_{core}).$$

Given that $\mathbb{E}[\epsilon_{spu}] = 0$ and $\mathbb{E}[\epsilon_{core}] = 0$, and considering $\mathbb{E}_{\mathcal{D}_{\text{train}}}[y] = \beta \mu_{core}$, the expectations become:

$$\mathbb{E}_{\mathcal{D}_{\text{train}}}[v_{spu}^* x_{spu}] = \frac{\eta_{core}^2}{\eta_{spu}^2 + \eta_{core}^2} \beta \mu_{core},$$

$$\mathbb{E}_{\mathcal{D}_{\text{test}}}[v_{core}^* x_{core}] = \frac{\eta_{spu}^2}{\eta_{spu}^2 + \eta_{core}^2} \beta \mu_{core}.$$

Moreover, with $\text{Var}[\epsilon_{spu}] = \eta_{spu}^2$, $\text{Var}[\epsilon_{core}] = \eta_{core}^2$, we find:

$$\text{Var}_{D_{tr}}[v_{spu}^* x_{spu}] = \frac{\eta_{core}^4}{(\eta_{spu}^2 + \eta_{core}^2)^2} (\eta_{spu}^2 + \beta \Sigma_{core} \beta^T),$$

$$\text{Var}_{D_{tr}}[v_{core}^* x_{core}] = \frac{\eta_{spu}^4}{(\eta_{spu}^2 + \eta_{core}^2)^2} (\eta_{core}^2 + \beta \Sigma_{core} \beta^T).$$

These derivations underpin the conclusion that when $\eta_{spu} < \eta_{core}$, the model is more inclined to rely on the spurious feature, as evidenced by $\mathbb{E}^2[c_{spu}] > \mathbb{E}^2[c_{core}]$ and $\text{Var}[c_{spu}] < \text{Var}[c_{core}]$, as posited in **Lemma 2**.

In proceeding with the proof of **Theorem 2** based on the intermediate steps and outcomes provided in **Lemma 2**, we aim to evaluate the probability that feature exhibits higher activation energy when it is more dependent on a spurious component compared to a core component.

**Proof of Theorem 2:** In the following, we denote

$$c_{core}^{(a)} = W_{ka} \cdot g_{core}^{(a)} x_{core}$$
$$c_{spu}^{(a)} = W_{ka} \cdot g_{spu}^{(a)} x_{spu}$$
$$c_{core}^{(b)} = W_{kb} \cdot g_{core}^{(b)} x_{core}$$
$$c_{spu}^{(b)} = W_{kb} \cdot g_{spu}^{(b)} x_{spu}$$

Notice that $\text{sign}(\mathbf{E}^{(ak)} - \mathbf{E}^{(bk)}) = -\text{sign}(c_{core}^{(a)} + c_{spu}^{(a)} - (c_{core}^{(b)} + c_{spu}^{(b)}))$. Based on the condition $W_{ka} \cdot g_{spu}^{(a)} x_{spu} > W_{kb} \cdot g_{spu}^{(b)} x_{spu}$, we can obtain $c_{spu}^{(a)} > c_{spu}^{(b)}$. Our focus is on a specific scenario characterized by a positive real number $t$, for which we examine the probability:

$$P(c_{core}^{(a)} + c_{spu}^{(a)} > c_{core}^{(b)} + c_{spu}^{(b)} \mid c_{spu}^{(a)} - c_{spu}^{(b)} = t) \tag{30}$$

This probability can be rewritten as:

$$P(c_{core}^{(b)} - c_{core}^{(a)} < t) \tag{31}$$

Using the computation results induced from **Lemma 2**:

$$c_{core} \sim N\left( \frac{\eta_{core}^2 \beta \mu_{core}}{\eta_{spu}^2 + \eta_{core}^2}, \frac{\eta_{spu}^4 (\eta_{core}^2 + \beta \Sigma_{core} \beta^T)}{(\eta_{spu}^2 + \eta_{core}^2)^2} \right) \tag{32}$$

Considering the independence of $c_{core}^{(b)}$ and $c_{core}^{(a)}$ given by $x_{core}$, their difference also follows a Gaussian distribution:

$$c_{core}^{(b)} - c_{core}^{(a)} \sim N(0, 2 \cdot \frac{\eta_{spu}^4(\eta_{core}^2 + \beta\Sigma_{core}\beta^T)}{(\eta_{spu}^2 + \eta_{core}^2)^2}) \tag{33}$$

Let Z be the standard normal variable, where $Z \sim N(0, 1)$, Then $P(c_{core}^{(b)} - c_{core}^{(a)} < t)$ can be translated to:

$$P(Z < \frac{t}{\sqrt{2} \cdot \left(\frac{\eta_{spu}^4(\eta_{core}^2 + \beta\Sigma_{core}\beta^T)}{(\eta_{spu}^2 + \eta_{core}^2)^2}\right)^{\frac{1}{2}}}) \tag{34}$$

Recall that our target is:

$$P(c_{core}^{(a)} + c_{spu}^{(a)} > c_{core}^{(b)} + c_{spu}^{(b)} \mid c_{spu}^{(a)} > c_{spu}^{(b)}) \tag{35}$$

To obtain this, we integrate from $t = 0$ to $t = 1$. Because $c_{core}$ is independent to $c_{spu}$ with known parameter, we can decompose the probability in our objective as follows:

$$\int_{t=0}^{\infty} P(c_{core}^{(b)} - c_{core}^{(a)} < t) \cdot P(c_{spu}^{(a)} - c_{spu}^{(b)} = t)dt$$

$$= \int_{t=0}^{\infty} P(Z < \frac{t}{\sqrt{2} \cdot \left(\frac{\eta_{core}^4(\eta_{spu}^2 + \beta\Sigma_{core}\beta^T)}{(\eta_{spu}^2 + \eta_{core}^2)^2}\right)^{\frac{1}{2}}}) P_Z(\frac{t}{\sqrt{2} \cdot \left(\frac{\eta_{spu}^4(\eta_{core}^2 + \beta\Sigma_{core}\beta^T)}{(\eta_{spu}^2 + \eta_{core}^2)^2}\right)^{\frac{1}{2}}})dt$$

We extract the residual expression as:

$$R(\eta_{spu}, \eta_{core}) = \int_{t=0}^{\infty} P(Z < \frac{t}{\sqrt{2} \cdot \left(\frac{\eta_{core}^4(\eta_{spu}^2 + \beta\Sigma_{core}\beta^T)}{(\eta_{spu}^2 + \eta_{core}^2)^2}\right)^{\frac{1}{2}}})$$

$$\left(P_Z(\frac{t}{\sqrt{2} \cdot \left(\frac{\eta_{spu}^4(\eta_{core}^2 + \beta\Sigma_{core}\beta^T)}{(\eta_{spu}^2 + \eta_{core}^2)^2}\right)^{\frac{1}{2}}}) - P_Z(\frac{t}{\sqrt{2} \cdot \left(\frac{\eta_{core}^4(\eta_{spu}^2 + \beta\Sigma_{core}\beta^T)}{(\eta_{spu}^2 + \eta_{core}^2)^2}\right)^{\frac{1}{2}}})\right) \quad dt$$

Therefore, the original expression can be induced as:

$$\int_{t=0}^{\infty} P(c_{core}^{(b)} - c_{core}^{(a)} < t) \cdot P(c_{spu}^{(a)} - c_{spu}^{(b)} = t)dt$$

$$= \int_{t=0}^{\infty} P(Z < \frac{t}{\sqrt{2} \cdot \left(\frac{\eta_{core}^4(\eta_{spu}^2 + \beta\Sigma_{core}\beta^T)}{(\eta_{spu}^2 + \eta_{core}^2)^2}\right)^{\frac{1}{2}}}) P_Z(\frac{t}{\sqrt{2} \cdot \left(\frac{\eta_{core}^4(\eta_{spu}^2 + \beta\Sigma_{core}\beta^T)}{(\eta_{spu}^2 + \eta_{core}^2)^2}\right)^{\frac{1}{2}}})dt + R(\eta_{spu}, \eta_{core})$$

$$= \quad 0.5 + R(\eta_{spu}, \eta_{core})$$

$$> \quad 0.5$$

In the third line above, we use the condition that $\eta_{core} < \eta_{spu}$ which is true when the provided training dataset is biased. Theorem 2 indicates that if a feature relies more heavily on a spurious feature, it is more likely to exhibit lower evidence energy.

## B   ALGORITHM

EvA is a two-stage methodology, comprising detection and reweighting phases. In the experimental implementation, assume the deep feature dimension is $m$. We erase $\lfloor \epsilon m \rfloor$ feature for each class. The measure of spuriousness is conducted using consistency in an open-box setting and evidence energy

in a closed-box setting. Following this, the `TopK` function is employed to select $\lfloor \epsilon m \rfloor$ pieces of feature with the highest spuriousness, as delineated in Algorithm 1.

During the reweighting stage, the weight corresponding to spurious feature is 'frozen' and assigned a value of 0 in the linear layer. The final step involves selecting the reweighted linear layer based on various erase ratio candidates, evaluating performance using the `Eval` function. The choice of evaluation metric in `Eval` depends on the specific hyperparameter goals outlined in Section 3.1. The entire algorithm is comprehensively illustrated in Algorithm2.

---

**Algorithm 1:** `Detect`$(\Phi, \mathcal{D}_{\text{train}}, \epsilon)$ / $(\Phi, \mathcal{D}_{\text{train}}, \mathcal{D}_{\text{ex}}, \epsilon)$

   **Input:**
   training dataset $\mathcal{D}_{\text{train}}$ with $K$ classes, extra dataset $\mathcal{D}_{\text{ex}}$ (optional);
   feature set $\Phi = \{\phi^{(i)} \mid i \in \{1, ..., m\}\}$ ;
   linear probing $h_\theta$, where: $\theta = \{(W_k, b_k) \in R^m \times R \mid k \in \{1, ..., K\}\}$
   **HParams:** erase ratio $\epsilon$ ;
   **Output:**
   spurious indicator $P = \{p_k \in \{0, 1\}^m\} \mid k = \{1, ..., K\}\}$

1  **for** $k = 1...K$ **do**
2     initialize $p_k = (0, ..., 0)$, $P = \{\}$;
3     **for** $i = 1...m$ **do**
4        $p_{ki} \leftarrow \mathbf{E}^{(ik)}/\mathbf{C}^{(ik)}$
5     **end**
6     $p_k^+ = \{p_{kj_1}, ..., p_{kj_{\lfloor \epsilon m \rfloor}}\} \leftarrow \text{TopK}(p_k, \lfloor \epsilon m \rfloor)$;
7     **for** $i = 1...m$ **do**
8        $p_{ki} \leftarrow \mathbb{1}(p_{ki} \in p_k^+)$
9     **end**
10    $P \leftarrow P \cup \{p_k\}$;
11 **end**
12 Return $P$

---

**Algorithm 2:** `Reweight`

   **Input:**
   training dataset $\mathcal{D}_{\text{train}}$, validation dataset $D_v$ with $K$ classes;
   feature set $\Phi = \{\phi^{(i)} \mid i \in \{1, ..., m\}\}$ ;
   linear probing $h_\theta$, where: $\theta = \{(W_k, b_k) \in R^m \times R \mid k \in \{1, ..., K\}\}$
   **HParams:** erase ratio candidates $\mathcal{E} = \{\epsilon_1, ..., \epsilon_T\}$ ;
   **Output:**
   re-weighted linear probing $h_{\theta*}$

1  Initialize $v^* = 0$
2  **for** $t = 1, ..., T$ **do**
3     $\{p_1, ..., p_K\} \leftarrow \text{Detect}(p, \mathcal{D}_{\text{train}}, \epsilon_t)$;
4     **for** $k = 1, ..., K$ **do**
5        **for** $i = 1, ..., m$ **do**
6           **if** $p_{ki} = 0$ **then**
7              `Freeze`$(W_{ki})$
8           **end**
9        **end**
10      $W_k = p_k W_k$;
11    **end**
12    $\theta_t \leftarrow \arg\min_\theta \text{Loss}(\Phi, \mathcal{D}_{\text{train}}, h_\theta)$
13    $v_t \leftarrow \text{Eval}(p, \mathcal{D}_{\text{unbiased}}, h_{\theta_t})$;
14    **if** $v_t > v^*$ **then**
15      $v^* = v_t$, $\theta^* = \theta_t$
16    **end**
17 **end**
18 Return $\theta^*$

---

## C  COMPARING METHODS

We provide additional details of comparing methods in this work based on two different categories: methods without training with extra data and methods require extra data for training. Then we discusses the main difference between our works and these comparing methods.

### C.1  WITHOUT EXTRA DATA AND GROUP LABEL

**LfF**. Nam et al. (2020) proposes a method to reduce bias in neural networks without requiring explicit labels or assumptions about the type of bias. They introduce a dual-network debiasing strategy: the first network is trained to amplify its biases, while the second network is trained using samples misclassified by the first, encouraging it to learn beyond the biases. This approach effectively mitigates bias in various datasets and often outperforms traditional methods that need more supervision. However, it trade-off computation efficiency for data-efficiency, i.e., reduces the reliance on group label but requires extensive computation on hyper-parameter selection over two neural networks together.

**JTT**. Liu et al. (2021) also proposes a two-stage method to improve worst-group accuracy without extensive group annotations. It first trains a standard ERM model and then re-trains by upweighting misclassified examples, thereby focusing on groups that ERM struggles with. JTT closes 75% of the performance gap between ERM and group DRO across various tasks with spurious correlations, while only needing minimal group annotations for hyperparameter tuning. However, it still requires retraining the based neural network to tune several hyperparameter.

**LWBC**. Kim et al. (2022) introduces a method to train debiased classifiers without using labels for spurious attributes. It uses a committee of classifiers to identify and assign higher weights to data that do not exhibit spurious correlations. The committee, which is intentionally biased and diverse, helps detect bias-conflicting samples by agreeing on their prediction difficulty. It learns alongside the main classifier, gradually becoming less biased and emphasizing harder examples over time. However, it also suffers from computation efficiency with training a commitee of classifiers.

**Debian**. Li et al. (2022) also addresses biases in deep image classifiers without relying on labels for protected attributes. DebiAN adopts similar idea with LfF (Nam et al., 2020) and the framework consists of two networks: a Discoverer that identifies unknown biases and a Classifier that unlearns them through alternate training. DebiAN effectively mitigates multiple biases simultaneously where most previous methods focus on single type of bias. But it also requires extensive computation to tune the hyperparameter.

**DivDis**. Lee et al. (2023) is another two-stages debiasing method without requiring label. DivDis first learns a diverse set of hypotheses using unlabeled test data, and then selects the most robust one with minimal extra data. It is shown to effectively identify hypotheses that rely on robust features. The computation cost can be reduced by training a singal model with different heads but still suffering from retraining the whole model during tuning stage.

**SiFER**. Tiwari & Shenoy (2023) focus on general simplicity bias rather than only spurious correlation and proposes feature sieve. During training iteration, they make assumption on simple features mostly can be extracted from lower layer and add additional penalty to make it harder to classify based on feature in the shallow layer. The method is effective in debiasing but lack computation efficiency in tuning a large number of hyperparameters such as designs of classifier for adding the panalty and the layer to append such classifier.

### C.2  WITH EXTRA DATA AND GROUP LABEL

**DFR**. Kirichenko et al. (2022) significant improves the efficiency from previous methods by applying last linear layer re-training to robustify the whole classifier. The method is simple but effective and the work provides insights that the model learns spurious feature as well as core feature during training but just biased to the spurious feature. And this bias can be suppressed by only reweighting the last linear layer by a less unbiased dataset. However, our analysis shows that DFR may be worse than the method without extra data in terms of data efficiency where there and  Ye et al. (2023) shows that DFR can only be effective when the extra training data is less unbiased, i.e., the noise exists in correlation with core feature and label should be less than the correlation with spurious feature. The

requirement for utilizing data for training makes it relies more on less biased data than those only use it for tuning hyperparameter.

### C.3 DIFFERENCES BETWEEN OUR WORK AND PREVIOUS METHODS

We notice that almost all existing methods without group label are two-stage where the first stage includes training and collecting information related to potential spurious correlation and removing it explicitly or indirectly in the second stage.

Commonly, the computation cost in the first stage is necessary to get the base model and debias on it. However, the computation cost at the second stage is usually very large to tune the hyperparameter by retraining another neural network or whole neural network, which makes these methods data-efficient but less computation-efficient.

On the contrary, DFR (Kirichenko et al., 2022) provides compute-efficient method by only reweighting linear layer in the second stage using unbiased dataset. However, its performance is only guaranteed when tuning with Worst Accuracy but not robust when there is no group label, weaken its data-efficiency.

EvA exploits the advantage from both sides. We observe that the deep spurious feature can be identified without group label but the activation distribution over training dataset (EvA-E) or validation dataset (EvA-C). This makes EvA as computation efficient as DFR (Kirichenko et al., 2022) and even more data-efficient than previous methods (Nam et al., 2020; Liu et al., 2021; Kim et al., 2022; Lee et al., 2023; Tiwari & Shenoy, 2023) not relying on group label as shown in Table 4.

Recently, several works, including AFR (Qiu et al., 2023), BAM (Li et al., 2023), and SELF (LaBonte et al., 2024), have also explored reweighting the last linear layer while relying on fewer or no additional annotations. However, these methods primarily focus on the data level, identifying failure cases during the first training stage and addressing model bias by leveraging these detected cases. In contrast, our approach focuses on debiasing the channel-level characteristics of the features. A promising direction for future research is to combine feature-level and data-level methods to further enhance reweighting strategies for feature representation.

## D DATASET DETAILS

In this section, we provide detailed group information and the specific size of each group in the datasets used for analysis and performance comparison. Each group is defined by a human-identified feature and a label from the dataset, represented as *(feature, label)*.

**CMNIST.** The Color-MNIST dataset Tiwari & Shenoy (2023) consists of four groups: *(Red, Zero)*, *(Red, One)*, *(Green, Zero)*, and *(Green, One)*. In the training dataset, the size of each group is (1024, 0, 0, 1024), respectively. In the validation and test datasets, the size of each group is (128, 128, 128, 128), making the datasets perfectly unbiased with respect to the defined feature. Consequently, models trained using ERM on the biased training dataset struggle to distinguish the digit label from the color attribute.

**BAR.** The BAR dataset Nam et al. (2020) is a standard benchmark for debiasing tasks. Although it does not provide explicit human-defined spurious features, it ensures that the background in the training images for each class is different from the background in the test images for the same class. The dataset contains images from six human action classes: "Climbing," "Diving," "Fishing," "Racing," "Throwing," and "Vaulting." It can be represented as 12 groups, with the first 6 corresponding to the training dataset and the remaining 6 to the test dataset. The sizes of the groups in the training dataset are (326, 520, 163, 336, 317, 279, 0, 0, 0, 0, 0, 0), while in the test dataset, the sizes are (0, 0, 0, 0, 0, 0, 105, 159, 42, 132, 85, 131), respectively. This dataset is challenging due to the complex spurious correlations and class imbalances. As shown in Table 6, the performance of debiasing methods on classes with fewer samples, such as "Fishing" and "Throwing," is often worse.

**Waterbirds.** The Waterbirds dataset Sagawa et al. (2019) is a benchmark for studying spurious correlations. It is synthesized from real-world data and includes four groups: *(land, landbird)*, *(land, waterbird)*, *(water, landbird)*, and *(water, waterbird)*. The sizes of each group in the training dataset are (3498, 184, 56, 1057), and in the validation dataset, the sizes are (467, 466, 133, 133), respectively.

| Method | Climbing | Diving | Fishing | Racing | Throwing | Vaulting | Average |
|--------|----------|--------|---------|--------|----------|----------|---------|
| LfF | $79.39_{\pm4.79}$ | $34.59_{\pm2.26}$ | $75.39_{\pm3.63}$ | $83.08_{\pm1.90}$ | $33.72_{\pm0.68}$ | $71.75_{\pm3.32}$ | $62.98_{\pm2.76}$ |
| DFR | $92.08^{\star}_{\pm1.18}$ | $93.55_{\pm2.47}$ | $70.39_{\pm8.51}$ | $92.97_{\pm3.64}$ | $67.92_{\pm4.19}$ | $91.33_{\pm2.43}$ | $84.70_{\pm1.67}$ |
| EvA-C | $90.47_{\pm2.79}$ | $94.78^{\star}_{\pm2.60}$ | $77.41^{\star}_{\pm3.06}$ | $97.19^{\star}_{\pm1.14}$ | $84.43^{\star}_{\pm6.73}$ | $92.72^{\star}_{\pm2.93}$ | $89.50^{\star}_{\pm1.50}$ |

Table 6: Top-1 Accuracy (with standard deviations) of LfF (Nam et al., 2020) and DFR (Kirichenko et al., 2022) with EvA-C (Ours) across different classes over BAR dataset. Asterisks ($\star$) indicate the best performance. Our method achieves much better results compared to previous methods.

| Base Model | Waterbirds | | CelebA | | FLOP for debiasing |
|------------|------------|------------|------------|------------|--------------------|
| Debiasing Method | Worst Acc | Mean Acc | Worst Acc | Mean Acc | |
| ResNet50 + JTT | **86.7%** | **93.3%** | 81.0% | 88.0% | $\approx 4 \times 10^9 \cdot$ cnms |
| ResNet50 + DivDis | 85.6% | - | 55.0% | - | $\approx 4 \times 10^9 \cdot$ cnms |
| ResNet50 + EvA-E | 86.6% | 92.5% | **82.7%** | **88.7%** | $\approx \mathbf{8 \times 10^3} \cdot$ cnms |

Table 7: Comparison of EvA-E (Ours) with two state-of-the-art methods, JTT Liu et al. (2021) and DivDis Lee et al. (2023) without using extra data for training. The performance here is all tuned with **Worst Acc**, i.e., the tuning dataset is with group label. The best performance is highlighted in **bold**. Our method achieves competitive worst-case and mean accuracy with six orders of magnitude less computational cost.

The minority groups in the training dataset are *(water, landbird)* and *(land, waterbird)*. Models trained with ERM on the entire training dataset tend to classify landbirds based on land backgrounds and waterbirds based on water backgrounds.

**CelebA Hair.** The CelebA Hair dataset (Liu et al., 2018) is another standard benchmark for studying spurious correlations and has a relatively larger size. It is a real-world dataset containing four groups: *(non-blond, woman)*, *(non-blond, man)*, *(blond, woman)*, and *(blond, man)*. The sizes of each group in the training dataset are (71629, 66874, 22880, 1387), and in the validation dataset, the sizes are (8535, 8276, 2874, 182), respectively. The minority group in the training dataset is *(blond, man)*. Models trained with ERM on the entire training dataset tend to incorrectly classify individuals with blond hair as women.

# E    ADDITIONAL EXPERIMENTAL RESULTS

## E.1    PERFORMANCE OVER EACH CLASSES ON BAR DATASET

We present additional experimental results on the BAR dataset with ResNet18, focusing on the mean and standard deviation of accuracy for each class. Our method, EvA-C, is compared against LfF (Nam et al., 2020) and DFR (Kirichenko et al., 2022), as shown in Table 6. EvA-C consistently outperforms the previous methods across most classes, demonstrating superior performance. Notably, it achieves significant improvements in the "Fishing" and "Throwing" classes, where the previous methods exhibited biases towards other classes.

## E.2    COMPARISON WITH STATE-OF-THE-ART METHODS ON RESNET-50 AND VIT

We extend our experiments analysis on the backbone ResNet-50 to compare our methods with three state-of-the-art methods inlcuding JTT (Liu et al., 2021), DivDis (Lee et al., 2023) and DFR (Kirichenko et al., 2022) to further demonstrate the computation and data efficiency of our methods while remaining competitive performance on both ideal and practical settings.

**Comparison between EvA-E and Methods without Extra Data for Training:** We compare the **Worst Accuracy** of EvA-E against JTT and DivDis on both unbiased and biased datasets, as shown in Table 4 in the main text. Our results demonstrate a significant improvement in accuracy when tuning data is biased, highlighting the practical applicability of our method in real-world scenarios. Following previous studies (Kirichenko et al., 2022; Sagawa et al., 2019), we also present results using the standard comparison method, where Worst Accuracy is measured with known group labels, as shown in Table 7.

In addition, we report the FLOPs in Table 7 to quantify the computational cost of debiasing the model, where $c$ denotes the constant computation overhead for updating the gradient of each parameter, $n$

| Conflicting Ratio | 50% | 40% | 30% | 20% |
|---|---|---|---|---|
| ResNet50 + EvA-C | 92.1 | 92.5 | 91.2 | 87.8 |
| ResNet50 + DFR | 90.7 | 89.0 | 86.6 | 83.2 |

Table 8: Comparison of the **Worst Acc** between EvA-C and DFR under different conflicting ratios on the Waterbirds Dataset. To make fair comparison where group label is not accessed in the real world, this result is obtained by tuning on **Mean Acc** rather than **Worst Acc** therefore the results is different from 92.9 in the original paper (Kirichenko et al., 2022) tuned with **Worst Acc**. Under the original setting using a "unbiased" dataset with a 50% conflicting ratio, our method still performs better than DFR. As the conflicting ratio decreases, i.e., the validation dataset is more imbalanced, EvA-C performs better than DFR.

represents the number of epochs, $m$ is the dataset size, and $s$ is the size of the hyperparameter search space. JTT requires retraining the entire model with a reweighted loss to select the appropriate weight for the error set, and DivDis also involves retraining the entire network to determine the weight of the mutual information loss during the diversification stage. In contrast, EvA-E only updates the linear layer for debiasing. For a ResNet50 model, the FLOPs for retraining the entire network are approximately $4 \times 10^9$, whereas updating the linear layer requires only about $8 \times 10^3$ FLOPs. Therefore, the computational cost of EvA-E is significantly lower than that of previous debiasing methods.

Considering the results in Table 4, our method not only significantly improves **computational efficiency** and **data efficiency** but also achieves competitive accuracy.

**Comparison between EvA-C and DFR with Extra Data for training:** Apart from the setting where extra data is not provided for training, we also extend the results on ResNet-50 on the settings where extra data is available, i.e., comparing EvA-C with DFR. To make fair comparison where group label is not accessed in the real world, this result is obtained by tuning on average accuracy, i.e., **Mean Acc** rather than worst accuracy, i.e., **Worst Acc** even when the conflicting ratio is 50% where it's unbiased. As demonstrated in Table 8, In the original setting, which has a 50% conflicting ratio, EvA-C outperforms Deep Feature Reweighting (DFR).

We further investigate how the performance of our methods improves with an increase in the conflicting ratio within the reweighting dataset. To manipulate the conflicting ratio, we selectively add non-conflicting samples and remove conflicting samples into the tuning dataset. For instance, in the Waterbirds dataset, which encompasses four groups - (water, water-birds), (water, land-birds), (land, water-birds), and (land, land-birds) - we adjust the conflicting ratio by adding samples from the (water, water-birds) and (land, land-birds) groups and eliminating samples from (water, land-birds) and (land, water-birds) to keep the size of tuning dataset as the same.

The data presented in Table 8 illustrate that our method significantly outperforms DFR as the conflicting ratio decreases. This result underscores the **data efficiency** of EvA-C, where EvA-C requires substantially fewer labeled data points to achieve relatively good results. This is in contrast to DFR, which is less robust to "biased" reweighting dataset.

**Comparison between EvA-E and DFR on ViT and ImageNet Background Challenge:** We additionally evaluated our method on ViT-B using ImageNet-9 [a], which involves background spurious correlations in real-world datasets. Our method improves the baseline model's accuracy from 87.9% to 88.7%, requiring no additional data or labels and only 10 minutes for hyperparameter tuning while DFR is harmful when performing on original dataset and only achieves 87.1%.

### E.3 ADDITIONAL ANALYSIS ON DETECTED SPURIOUS FEATURE

**Activation distribution of detected spurious feature:** We conduct a comparative analysis of the activation distribution between potential spurious feature, identified by evidence energy, and the remaining feature in the BAR dataset. This comparison is visually represented in Figure 4. Notably, the activation distribution disparity between the training and validation datasets is significantly more pronounced for the detected spurious feature than for the remaining feature. This further echos Theorem 2 that if the feature exhibits higher evidence energy, it's more likely to have higher consistency. And it's consistent with the empirical analysis on Section 4.2 Evidence Energy & Consistency.

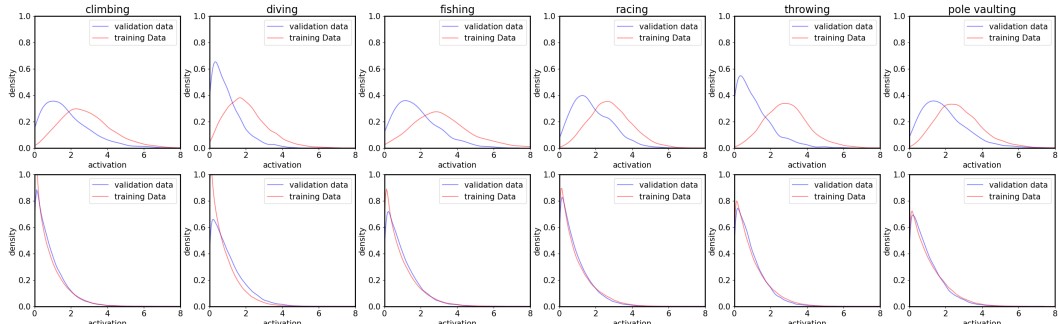

Figure 4: Activation distribution of potential spurious feature (the first row), identified by evidence energy, compared to the remaining feature (the second row). The potential spurious feature exhibits a more marked difference in activation distribution between the training and validation datasets than the remaining feature, which is categorized as core feature.

$$E_{core} = 0.0029 \quad C_{core} = -0.192 \quad E_{spu} = -0.0019 \quad C_{spu} = -0.339$$

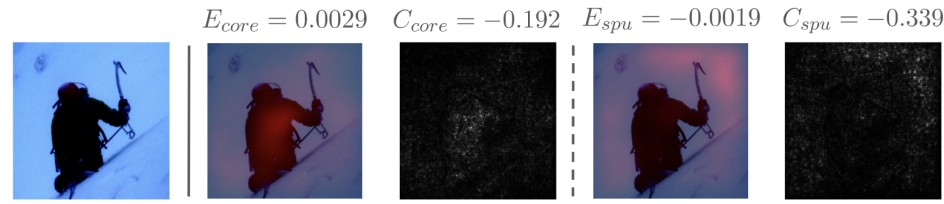

(a) Input image    (b) Saliency map of core feature    (c) Saliency map of spurious feature

Figure 5: Saliency map of the activation on two feature annotated with their evidence energy (E) and consistency (C) for class *"climbing"* with respect to the input image (a) from BAR dataset (Nam et al., 2020). For action recognition task, background is considered as spurious feature while the human body is considered as core feature. (b) The feature with higher consistency and evidence energy, which is identified as core feature, focus more on the human body; while (c) the other feature detected as spurious feature by EvA with lower evidence energy and consistency, is attributed to the background.

**Saliency map of detected spurious feature**: We further analyze the feature detected by both EvA-E and EvA-C by visualizing their corresponding saliency maps. We use the BAR dataset for this analysis, as it represents a real-world scenario where the distinction between core and spurious features is clear. In the BAR dataset, which is designed for action recognition, the background is considered a spurious feature, while the human body region is regarded as the core feature. For instance, in the training set, the action "climbing" is predominantly associated with a normal mountain background rather than a snow-covered one.

To indicate which input component used for a specific feature, we use Integrated Gradient (Sundararajan et al., 2017) to compute the saliency map of the value of feature with respect to the input as Figure 5 shows. The core feature identified by both EvA-E and EvA-C assigns higher importance on the human body area while the detected spurious feature assigns higher importance on the background, which is also aligned with human intuition.

## E.4 ADDITIONAL ABLATION STUDIES

**EvA v.s. Retrain Linear Layer with Lasso Regression:** Notice that the implementation of DFR implicitly considering tuning the weight decay which is the strength of L1 Norm. Therefore, the comparison between EvA and Lasso Regression on last linear layer is directly reflected when comparing EvA and DFR.

**Other spuriousness indicator:** In a closed-box setting, where additional data including spurious feature information is unavailable, we extend our ablation study to include evidence energy alongside other indicators like *strength* and *instability*.

| Class | climbing | diving | fishing | racing | throwing | pole vaulting |
|---|---|---|---|---|---|---|
| Evidence Energy | **0.46** | **0.34** | **0.73** | **0.69** | **0.70** | **0.46** |
| Strength | 0.44 | 0.29 | 0.62 | 0.64 | 0.63 | 0.42 |
| Instability | 0.17 | 0.20 | 0.37 | 0.27 | 0.37 | 0.16 |

Table 9: Pearson correlation between three indices—strength, instability, and evidence energy—and their consistency on the BAR dataset. It is observed that evidence energy has a higher correlation with consistency compared to the other two indices.

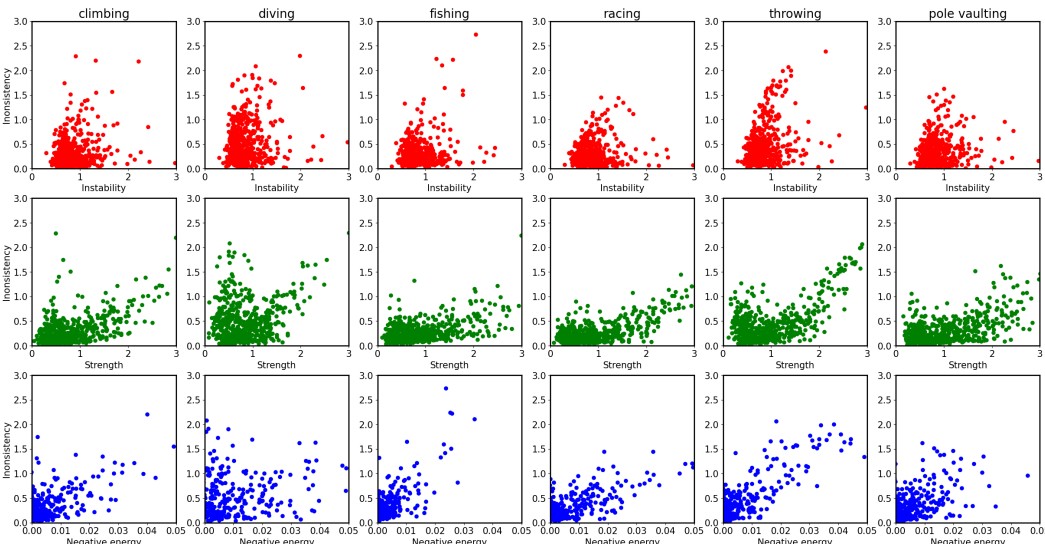

Figure 6: This figure illustrates the correlation between three indices—instability, strength, and negative evidence energy—and inconsistency across six classes in the BAR dataset. In the subplots, each point represents a data point from a specific class. The rows, ordered from top to bottom, correspond to instability, strength, and negative evidence energy, respectively. The analysis reveals that both negative evidence energy and strength exhibit a positive correlation with inconsistency, while instability does not display a similar correlation trend.

Concretely, the mean of an feature's activation distribution is termed its strength, while the variance is defined as instability. For a given feature $\phi^{(i)}$, a class $k$ and a dataset $D$, *strength* is formulated as:

$$S(\phi^{(i)}, k, D) = \mathbb{E}_{x \in D_k}[\phi^{(i)}(x)]$$
$$\approx \frac{1}{|D_k|} \sum_{x \in D_k} \phi^{(i)}(x) \tag{36}$$

Similarly, *instability* is defined as:

$$I(\phi^{(i)}, k, D) = \mathbb{E}_{x \in D_k}[(\phi^{(i)}(x) - S(\phi^{(i)}, k, D))^2]$$
$$\approx \frac{1}{|D|} \sum_{x \in D_k} (\phi^{(i)}(x) - S(\phi^{(i)}, k, D))^2 \tag{37}$$

where $D_k = \{x \mid (x, y) \in D, y = k\}$ represents the subset of data with the ground truth label $k$.

In comparison to strength and instability, evidence energy demonstrates a stronger correlation with consistency, as illustrated in Table9 and Figure6. This observation aligns with our findings from the BAR dataset. Experimentation with variations of EvA, including EvA-S (erasure based on strength) and EvA-I (erasure based on instability), yielded accuracies of $71.81\%$ and $64.81\%$ respectively. These results are lower than the $73.70\%$ accuracy achieved by EvA-E. Interestingly, strength is relatively good as well for detecting spurious feature. This may be because that the strength indicates the level of reliance on a certain input component as well.

