# OpenReview forum: "EvA: Erasing Spurious Correlations with Activations"
_ICLR.cc/2025/Conference — ICLR 2025 Poster_

### Official Review · Reviewer_iH3N · 2024-11-02

**Soundness:** 2
**Presentation:** 3
**Contribution:** 2
**Rating:** 6
**Confidence:** 3

**Summary:**

This paper proposes EVA, a method for addressing spurious correlations in datasets. It identifies spurious features using a consistency measure and the evidence energy measure in scenarios with and without unbiased datasets, respectively. Theoretical analyses are provided regarding the relationship between these two measures and a feature’s spuriousness. Experiments are conducted to demonstrate the method's effectiveness.

**Strengths:**

- The method is well-motivated, and I appreciate its simplicity (in terms of both understanding and computation).
- Providing theoretical analysis adds a more principled foundation to the two measures.
- There are experiments showing that the erased features are indeed the spurious ones on CMNIST, and the correlation between evidence energy and consistency, which further supports that the method works as expected.

**Weaknesses:**

- Since the original DFR paper uses ResNet50 as the architecture, it would be a good sanity check to compare results in that same setting. The authors provided results for ResNet50 in Appendix E.2, but they mention that, to mimic real-world scenarios, they tuned hyperparameters based on Mean Accuracy instead of Worst Accuracy. This might be debatable, as the setting where DFR is applicable assumes an unbiased validation set. Additionally, I wonder if the setup is consistent between ResNet18 in the main paper and ResNet50 in the appendix for the Waterbirds dataset. Could you clarify whether hyperparameters for ResNet18 were tuned based on Mean Accuracy or Worst Accuracy?
- Overall, since most existing papers evaluate their methods on ResNet50 while this paper primarily uses ResNet18 (this makes the numbers not comparable to those reported in other papers; in general, the numbers are much lower for all methods than those in the literature potentially due to the architecture), and some baselines (e.g., JTT, SSA [1], AFR [2], and [3]) are not included in the main table, Table 5, it is hard to compare the proposed method to the state of the art and to assess whether it truly stands out among existing techniques. For example, I wonder if the proposed method can outperform JTT on the Waterbirds dataset when ResNet50 is used, which has a reported worst-group accuracy of 86.7%, a relatively high value compared to the numbers in this paper obtained on ResNet18.
- Perhaps a detailed discussion and comparison with [1] is necessary, given that it also achieves the same goal: lightweight reweighting of the last layer without group labels.


[1] Qiu, Shikai, et al. "Simple and fast group robustness by automatic feature reweighting." International Conference on Machine Learning. PMLR, 2023.

[2] Nam, Junhyun, et al. "Spread spurious attribute: Improving worst-group accuracy with spurious attribute estimation." arXiv preprint arXiv:2204.02070 (2022).

[3] Liu, Sheng, et al. "Avoiding spurious correlations via logit correction." arXiv preprint arXiv:2212.01433 (2022).

**Questions:**

See the questions raised in the Weaknesses section.

---

> ### Author Response · Authors · 2024-11-25
>
> > W1. Since the original DFR paper uses ResNet50 as the architecture, it would be a good sanity check to compare results in that same setting. The authors provided results for ResNet50 in Appendix E.2, but they mention that, to mimic real-world scenarios, they tuned hyperparameters based on Mean Accuracy instead of Worst Accuracy. This might be debatable, as the setting where DFR is applicable assumes an unbiased validation set. Additionally, I wonder if the setup is consistent between ResNet18 in the main paper and ResNet50 in the appendix for the Waterbirds dataset. Could you clarify whether hyperparameters for ResNet18 were tuned based on Mean Accuracy or Worst Accuracy?
>
> Thank you for the question. For both ResNet18 and ResNet50 in Tables 5 and 7, hyperparameters were tuned based on Worst Accuracy. Thus, the comparison with DFR is fair, as both EvA-C and DFR are evaluated using an unbiased validation set in Table 5. But we provide additional results to empirically verify that EvA-E does not rely on Worst Acc as shown in Table. 4.
>
> > W2. Overall, since most existing papers evaluate their methods on ResNet50 while this paper primarily uses ResNet18 (this makes the numbers not comparable to those reported in other papers; in general, the numbers are much lower for all methods than those in the literature potentially due to the architecture), and some baselines (e.g., JTT, SSA [1], AFR [2], and [3]) are not included in the main table, Table 5, it is hard to compare the proposed method to the state of the art and to assess whether it truly stands out among existing techniques. For example, I wonder if the proposed method can outperform JTT on the Waterbirds dataset when ResNet50 is used, which has a reported worst-group accuracy of 86.7%, a relatively high value compared to the numbers in this paper obtained on ResNet18.
>
> Thank you for pointing this out. Table 4 presents results using the ResNet50 architecture, as described in Appendix E.2. JTT achieves a worst-group accuracy of 86.7\%, while EvA-E achieves 86.6\% when tuned for Worst Accuracy. This demonstrates that, even with group annotations available, our method is competitive with prior approaches. Furthermore, when group annotations are unavailable, our method outperforms JTT by 23.3\%. We will clarify this in the revision.
>
> > W3. Perhaps a detailed discussion and comparison with [1] is necessary, given that it also achieves the same goal: lightweight reweighting of the last layer without group labels.
>
> Thank you for highlighting this relevant work. The main contribution of AFR lies in its construction of a targeted reweighting dataset during the initial model training phase, which requires knowledge of the training details from the first stage. This enables AFR to achieve high worst-group accuracy. In contrast, our method does not rely on prior training information and instead identifies spuriousness at the feature level. Notably, our method could potentially complement AFR, enhancing its performance. We will include a detailed comparison in the revision and explore this integration in future work.

---

### Official Review · Reviewer_rgWi · 2024-11-04

**Soundness:** 2
**Presentation:** 3
**Contribution:** 2
**Rating:** 6
**Confidence:** 4

**Summary:**

Existing methods on mitigating spurious correlations typically requires unbiased data and multiple rounds of retraining. The paper proposes a computational- and data-efficient pipeline that detects spurious features before the last linear layer. When there is no unbiased datasets, the contribution of each feature to the network’s prediction
confidence is exploited for detection.  If the unbiased data is available, the consistency of penultimate activations between spurious and unbiased datasets is used for detecting spurious features.  Through channel-based erasure and re-weighting on the final linear layer, reliance on spurious correlations can be effectively mitigated.

**Strengths:**

- Two practical methods, namely EvA-C and EvA-E, are proposed for detecting and mitigating spurious features in the penultimate activations, depending on the existence of an unbiased dataset. The two proposed methods are both efficient and effective in mitigating spurious correlations.

- The two metrics originate from the spurious feature detection methods can be used to measure the spuriousness of features from the penultimate layer of a model. These metrics are also useful to validate the effectiveness of spurious correlation mitigation methods.

**Weaknesses:**

- Contrary to the claim in Line 93, the performance of the proposed method is not the state-of-the-art in comparison with some methods [1,2]. For example, with ResNet50 as the backbone, EvA-E has a worst-group accuracy of 86.6% (Table 7), while [2] achieves 89.1%.

- The unbiased dataset $\mathcal{D}\_{\text{unbiased}}$ is first introduced in Line 156; however, it is unclear what $\mathcal{D}\_{\text{unbiased}}$ can be considered as "unbiased". Based on the descriptions in Line 230 and Line 312, $\mathcal{D}\_{\text{unbiased}}$ is selected from the validation set. In some datasets, such as CelebA, the validation set also has gender bias. If $\mathcal{D}\_{\text{unbiased}}$ is defined as a set of group-balanced data where each class of samples distribute equally across different spurious features, then group annotations are required to obtain $\mathcal{D}\_{\text{unbiased}}$, contrary to the claim in Line 50-51. Further clarification on this point would be beneficial.

- In Eq. (4), why the unknown test distribution $\phi_{\text{test}}^{ik}$ can be approximated via $\phi_{\text{unbiased}}^{ik}$? In some datasets such as CelebA, the test set is also biased. It would be helpful to further clarify this point.

[1] LaBonte et al., Towards last-layer retraining for group robustness with fewer annotations, NIPS, 2023.\
[2] Li et al., Bias Amplification Enhances Minority Group Performance, TMLR, 2024.

**Questions:**

- Why spurious features often lead to high confidence predictions?
- Can the proposed method used for other model architectures beyond convolutional neural networks?

---

> ### Author Response · Authors · 2024-11-25
>
> > W1. Contrary to the claim in Line 93, the performance of the proposed method is not the state-of-the-art in comparison with some methods [1,2]. For example, with ResNet50 as the backbone, EvA-E has a worst-group accuracy of 86.6% (Table 7), while [2] achieves 89.1%.
>
> Thank you for providing additional references, especially recent works. In many cases our method is competitive with [1] and [2]. For instance, on CelebA, for worst accuracy, EvA-E achieves 82.7%, compared to 80.1% for [2], and 73.3\% for [1] when no extra label is available. Additionally, our method is competitive, particularly when considering computational and data efficiency. We will incorporate a discussion of these methods in the revision.
>
> > W2. The unbiased dataset $D_{unbiased}$  is first introduced in Line 156; however, it is unclear what $D_{unbiased}$ can be considered as "unbiased". Based on the descriptions in Line 230 and Line 312, $D_{unbiased}$ is selected from the validation set. In some datasets, such as CelebA, the validation set also has gender bias. If $D_{unbiased}$ is defined as a set of group-balanced data where each class of samples distribute equally across different spurious features, then group annotations are required to obtain $D_{unbiased}$, contrary to the claim in Line 50-51. Further clarification on this point would be beneficial.
>
> Similar to DFR  (Kirichenko et al., 2022), we refer to $D_{unbiased}$ as a set of group-balanced data. This is not contrary to the claim in line 50-51, since EvA-E does not require annotations and only EvA-C requires a small set of group-balanced data.
>
> > W3 In Eq. (4), why the unknown test distribution $\phi_{test}^{ik}$ can be approximated via $\phi_{unbiased}^{ik}$? In some datasets such as CelebA, the test set is also biased. It would be helpful to further clarify this point.
>
> Thank you for this correction. To clarify, the unknown test distribution refers to a perfect group-balanced dataset with unlimited samples, as the goal is to improve worst-group and mean accuracies across all groups. We will revise the notation in the paper to make this clearer.
>
> > Q1. Why spurious features often lead to high confidence predictions?
>
> Thank you for the question. This phenomenon, referred to as simplicity bias, has been documented in prior works, such as Tiwari & Shenoy (2023) and Xu et al. (2019). Models tend to prefer features with lower noise, which are often spurious. FTT (Ye et al., 2023) further supports this, showing that models prioritize features that are easier to learn. In Section 3.4, Theorem 2 formally demonstrates that when spurious features exhibit lower noise than core features, the evidence energy (and thus confidence) is higher. Intuitively, models are more confident on the features that are more prevalent and easier to learn from the training data.
>
> > Q2. Can the proposed method be used for other model architectures beyond convolutional neural networks?
>
> We have evaluated our method on ViT-B using ImageNet-9 [a], which involves background spurious correlations in real-world datasets. Our method improves the baseline model’s accuracy from 87.9% to 88.7%, requiring no additional data or labels and only 10 minutes for hyperparameter tuning while DFR is harmful when performing on original dataset and only achieves 87.1\%. We will include further results in the revision.
>
> [a] Xiao, K., Engstrom, L., Ilyas, A., & Madry, A. (2020). Noise or Signal: The Role of Image Backgrounds in Object Recognition. arXiv preprint arXiv:2006.09994.

---

> > ### Comment · Reviewer_rgWi · 2024-11-28
> >
> > I thank the authors for their detailed feedback.
> > Most of my concerns are addressed. Please add the comparison with more recent works in the paper and correct the notation.
> > Some additional comments:
> > - I still have some concerns about the experimental results. In Table 1, why DFR's performance on the Waterbirds dataset is much lower than the that reported in the original paper?
> > - For EvA-E, it requires a threshold to erase certain weights. Thus, the threshold is very important in determining the effectiveness of the method. How is the threshold determined? The validation data used for selecting the threshold may be as biased as the training data, such as the data in the CelebA dataset.
> > - For EvA-C, could you give some insights into why this method performs better than DFR? Both methods use an unbiased dataset for retraining.

---

> > > ### Author Response · Authors · 2024-11-28
> > >
> > > Thanks for your constructive feedback. We will follow your suggestions to add the comparison with more recent works in the paper and correct the notation. Below we addressed the concern one by one.
> > >
> > > > I still have some concerns about the experimental results. In Table 1, why DFR's performance on the Waterbirds dataset is much lower than the that reported in the original paper?
> > >
> > > If we understand correctly, you are referring to Table 5, which presents the main experimental results for ResNet-18.  Table 1 provides dataset statistics.
> > >
> > > In the original paper, the performance of DFR on the Waterbirds dataset was reported using the ResNet-50 backbone. Our Table 5 shows results with ResNet-18 so it does not match the results reported in the original paper.
> > >
> > > If we compare DFR to EvA-C with ResNet-50 on a sufficiently large unbiased dataset tuned for worst accuracy, the two methods are equivalent (92.9% vs. 92.9%).  Table 8 compares the two methods based on mean accuracy on Waterbird, for which EvA-C outperforms DFR (92.1\% vs 90.7\%), suggesting that EvA-C is more data-efficient than DFR. We further explain this in the response to your third question.
> > >
> > > > For EvA-E, it requires a threshold to erase certain weights. Thus, the threshold is very important in determining the effectiveness of the method. How is the threshold determined? The validation data used for selecting the threshold may be as biased as the training data, such as the data in the CelebA dataset.
> > >
> > > Thank you for the question. The threshold for EvA-E is determined using mean accuracy over validation data, even when the validation dataset is biased. Our method remains robust despite the bias.
> > >
> > > Intuitively, the trend in mean accuracy aligns with worst-case accuracy across different erase ratios on the biased dataset. This may be because most errors in the original model stem from the minority group. For instance, in the ResNet-50 and CelebA validation dataset, 92% of misclassified samples belong to the (blond, man) group. Proper regularization, such as limiting reliance on spurious features in EvA-E, minimally impacts the majority group's accuracy. Thus, improvements in mean accuracy for the biased validation dataset likely reflect gains in the minority group.
> > >
> > > Therefore, as shown in Table 4 (with ResNet-50), EvA-E achieves 85.8% accuracy when being tuned on mean accuracy of the biased validation dataset, only slightly lower than the 86.6% achieved with unbiased validation data.
> > >
> > > > For EvA-C, could you give some insights into why this method performs better than DFR? Both methods use an unbiased dataset for retraining.
> > >
> > > The primary reason EvA-C outperforms DFR lies in its ability to effectively handle the limited size of unbiased datasets, which are often much smaller than biased datasets in practical scenarios. DFR’s approach of fine-tuning with a small unbiased dataset may not adequately mitigate spurious correlations (e.g., assigning lower weights to spurious features).  In contrast, EvA-C directly eliminates potential spurious features, enabling reweighting to focus exclusively on the remaining core features. This approach is particularly effective when the unbiased dataset is limited.
> > >
> > > Figure 2(c) and Table 8 highlight the comparison between EvA-C and DFR across varying dataset sizes and conflicting ratios in the additional validation data. Notably, EvA-C demonstrates superior robustness, even when the unbiased dataset is smaller or less comprehensive.
> > >
> > > We sincerely hope that our response has fully addressed your concerns and you could reconsider the rating accordingly.

---

> > > > ### Comment · Reviewer_rgWi · 2024-12-03
> > > >
> > > > Thank you for the additional clarification. I am satisfied with the response and have accordingly increased my score.

---

### Official Review · Reviewer_Feuq · 2024-11-04

**Soundness:** 3
**Presentation:** 3
**Contribution:** 2
**Rating:** 6
**Confidence:** 3

**Summary:**

The paper introduces a method to detect and mitigate spurious correlations learned by an image classifier. They present two measures to identify channels in the penultimate layer of the model which correspond to such spurious correlations, one of them requiring an additional unbiased dataset (“consistency”) while the other can be computed on the the training data (“evidence energy”). After identifying a set of spurious channels, the corresponding weights in the last layer are set to zero and the remaining weights (of the last layer) are retrained.

**Strengths:**

- computational cost are lower compared to similar methods that do not require unbiased data
- their mitigation outperforms other methods on several common spurious correlation benchmarks
- for the variant including unbiased data, the introduced method requires less samples for the same accuracy
- the procedure seems to be well motivated including theoretical analysis

**Weaknesses:**

- the experiments establishing the correlation between their two measures are based only on one class of one of the datasets
- experiments are only based on small toy settings with strong spurious correlations
- the method is strongly inspired by OOD methods, but apart from a brief mention in the conclusion, this is not discussed in the main paper

minor:  fontsize in Fig. 1, Tab. 3, Tab. 4 are too small, the definition of “energy” actually corresponds to the “free energy” which might be confusing

**Questions:**

In Table 5, how many runs were performed to compute mean and std, and was the same erase ratio used in all of the EvA runs?

Section 4.4 (and Fig. 3 c) discuss the effect of the erase ratio on the worst accuracy. How does this hyperparameter influence the mean accuracy?

In Table 7, mean acc and worst acc are reported for CelebA but unbiased/conflicting acc in Table 5. The results on Waterbirds suggest that your method results in a better worst acc but worse mean acc for the ResNet-50 than for the ResNet-18: Does this pattern hold on CelebA?

Given the computational efficiency and no requirement for unbiased data or annotations of the spurious correlations, it should be quite feasible to evaluate EvA-E on ImageNet scale spurious correlations benchmarks (e.g. [1],[2]). In these more realistic settings, the distinction between spurious and core features become more complex. Can you provide empirical evidence or arguments that your evidence energy measure can still be used for detection and mitigation in such settings?


[1] Singla et al., Salient ImageNet: How to discover spurious features in Deep Learning?

[2] Neuhaus et al., Spurious Features Everywhere -- Large-Scale Detection of Harmful Spurious Features in ImageNet

---

> ### Author Response · Authors · 2024-11-25
>
> > W1. the experiments establishing the correlation between their two measures are based only on one class of one of the datasets
>
> Thank you for your comment. We respectfully disagree; our results demonstrate the correlation across all six classes of the BAR dataset, providing a broader evaluation on real-world images compared to other synthesized datasets with only two classes such as Waterbirds as suggested by [a], as shown in Appendix Figure 6.
>
> > W2. experiments are only based on small toy settings with strong spurious correlations
>
> Thank you for your feedback. Our experimental settings are consistent with established benchmarks used in prior works, such as JTT (Liu et al., 2021), DivDis (Lee et al., 2023), and FTT (Ye et al., 2023), all of which utilize the same datasets. Furthermore, the BAR dataset [a] was specifically introduced to address the lack of real-world benchmarks with complex spurious correlations. To address your concern, we have also included experiments on ImageNet-9 for additional analysis, as detailed in our response to Q4.
>
> > W3. the method is strongly inspired by OOD methods, but apart from a brief mention in the conclusion, this is not discussed in the main paper
>
> Thank you for highlighting this point. Indeed, our work is inspired by OOD methods.
> To clarify, the evidence energy is fundamentally motivated by the observation on lines 172–173: “based on the observation that there is a high probability of spurious correlations for network predictions with high confidence or prediction logits.” The first paragraph of the conclusion will be revised to more accurately reflect this motivation as follows:
>
> “To address spurious correlation, we focus on each channel of the deep feature and formally define spurious indicators—using consistency when unbiased data is available and evidence energy otherwise. Our findings show that erasing potential spurious features and reweighting core features can mitigate spurious correlations, regardless of additional information availability.”
>
> While our method and OOD approaches are related through free energy, we have focused on the energy perspective. We will correct this in the revision and add more discussion with OOD works.
>
> > Q1. In Table 5, how many runs were performed to compute mean and std, and was the same erase ratio used in all of the EvA runs?
>
> All results are reported as the mean and standard deviation over ten runs (lines 280–281). The erase ratio is tuned based on the validation dataset. Empirically the erase ratio selected is almost the same across different runs given the same dataset and backbone. For example, on ResNet-18 with Waterbirds debiased by EvA, the erase ratio of EvA-E is stable at 50\%.
>
> > Q2. Section 4.4 (and Fig. 3 c) discuss the effect of the erase ratio on the worst accuracy. How does this hyperparameter influence the mean accuracy?
>
> The mean accuracy remains stable initially but drops significantly as the erase ratio increases. For EvA-E on the Waterbirds dataset, we report the mean accuracy using the same model as in Figure 3c. The accuracy values are 97.3%, 97.5%, 96.9%, 97.1%, 96.9%, 94.2%, 90.7%, and 84.1% for progressively higher erase ratios. Notably, a clear decline in mean accuracy is observed beyond a certain threshold of the erase ratio. This observation will be included in the revised version.
>
> > Q3. In Table 7, mean acc and worst acc are reported for CelebA but unbiased/conflicting acc in Table 5. The results on Waterbirds suggest that your method results in a better worst acc but worse mean acc for the ResNet-50 than for the ResNet-18: Does this pattern hold on CelebA?
>
> Thank you for your insightful observation. Regarding the evaluation metrics in Table 7, they are consistent with the JTT paper [b]. For the unbiased/conflicting accuracy, the metrics align with the SiFER paper [c] to ensure comparability with the results in the original publications.
>
> In terms of worst-case accuracy and mean accuracy, this pattern is clearly observed in the CelebA dataset. ResNet-50 achieves higher worst-case accuracy but similar or lower mean accuracy compared to ResNet-18. This occurs because the tuning objective is explicitly optimized for worst-case accuracy, while the mean accuracy on both the Waterbirds and CelebA datasets is inherently biased. Consequently, the mean accuracy of ResNet-50 can only be comparable to or lower than that of ResNet-18.
>
> [a] Nam, J., Cha, H., Ahn, S., Lee, J., & Shin, J. (2020).
> [b] Liu, E. Z., e.t., al. JTT (2023).
> [c] Tiwari, R., & Shenoy, P. (2024).

---

> ### Author Response · Authors · 2024-12-03
>
> > Q4... it should be quite feasible to evaluate EvA-E on ImageNet scale spurious correlations benchmarks (e.g. [1],[2]) ... Can you provide empirical evidence or arguments that your evidence energy measure can still be used for detection and mitigation in such settings?
>
> We clarify that both BAR and CelebA involve real images without any toy data, and the spurious features in BAR are particularly complex, as the backgrounds are varied without specific patterns. To address your concern, we provide results on the ImageNet-9 [d] Background Challenge, a common benchmark for spurious correlations in real-world datasets. Using ViT-B, our method improves the baseline accuracy from 87.9\% to 88.7\% without additional data or labels, requiring only 10 minutes for hyperparameter search. In contrast, DFR reduces accuracy to 87.1\% when only original data is available. These results support the effectiveness of our approach in more realistic settings.
>
> [d] Xiao, K., Engstrom, L., Ilyas, A., & Madry, A. (2020). Noise or signal: The role of image backgrounds in object recognition. arXiv preprint arXiv:2006.09994.

---

> > ### Author Response · Authors · 2024-12-03
> >
> > Dear reviewer, thank you again for your thoughtful and detailed feedback on our submission. We truly appreciate the effort you have taken to review our work and provide constructive comments. We kindly ask you to consider whether our response addresses your concerns and, if so, whether it might warrant a reassessment of the score. Of course, we fully respect your judgment and appreciate your review.

---

> > > ### Comment · Reviewer_Feuq · 2024-12-03
> > >
> > > Thanks to the authors for the detailed rebuttal. Most of my concerns were adressed. I still have doubts regarding more realistic large scale settings which is not covered properly by common benchmarks. However, in total, the contributions of the paper outweigh this and I adjusted my score accordingly.

---

### Official Review · Reviewer_Jy78 · 2024-11-04

**Soundness:** 3
**Presentation:** 3
**Contribution:** 3
**Rating:** 8
**Confidence:** 4

**Summary:**

This paper presents an approach to mitigate biases learned through spurious correlations by erasing dimensions in the embedding space that are more likely to be associated with spurious features. The authors propose two metrics - namely consistency and evidence energy, for cases when additional unbiased data respectively are and are not available for training, which are used to identify spurious dimensions in the embedding space. Experiments reveal significant improvements over state-of-the-art bias mitigation approaches mainly in the absence of unbiased training data, but also to some extent when it is available.

**Strengths:**

1. The authors propose a post-hoc method for developing robustness to spurious correlations, which does not involve any additional training / heavy computational workload, which is a big positive in my opinion. The method operates by identifying specific dimensions in the embedding space that satisfy certain threshold criteria for some forms of spuriosity metrics proposed by the authors, dropping them from the data representations, and solving a logistic regression from this updated embedding space to the label space. None of these operations seem to be computationally expensive.

2. The authors propose ways to measure the degree of spuriosity of a specific dimension, when additional unbiased samples both are and are not available. This is a useful contribution which may also be applicable in other scenarios beyond bias mitigation.

3. The experimental results are strong. Performance improvements over state-of-the-art are quite significant on standard benchmarks, with minimal extra computational overhead.

**Weaknesses:**

1. The calculation of both consistency (eq. 4) and evidence energy (eq 7) are done independently for each feature dimension $i$. However, it is well known that input features do not necessarily map neatly onto independent dimensions in the feature space [a, b]. Given this, the feature erasure in eq. 9 should technically not be achieving what it claims to achieve. By turning full dimensions on/off, it could be - (i) partially turning off some core features that could be in superposition with a spurious feature; and (ii) not fully turning off spurious features, since it may be in superposition with some core feature dimension which is not turned off.

2. In computing $C_{(ik)}$ in eq. 4, there would be significant disparity in the distribution of the features stemming from sampling bias, since the number of unbiased samples is typically << the number of biased samples in practical datasets, which itself could skew the value of $d$, not providing a faithful measure of consistency. How do the authors account for this imbalance when applying the distance metric $d$?

3. Since the idea of EvA is essentially to remove dimensions from the embedding space that correspond to spurious features, an ablation for what constitutes the best approach for selecting the subspace to drop is necessary. EvA employs thresholding on consistency and evidence energy to select spurious subspaces. However, whether consistency and evidence energy are really the best metric to look at when it comes to performing this selection needs to be either theoretically or empirically established. For this, one empirical baseline could be comparison with random feature drop out in the embedding space. I would suggest the authors to explore this in further detail and consider developing such baselines to establish the optimality of their specific feature erasure mechanisms.

4. Table 5 shows that the gains from EvA are more significant in the absence of extra unbiased training data. The authors should provide a discussion on why their method might have a stronger advantage over SOTA when unbiased training data is not present.

5. When extra training data is not available, how does one perform the hyperparameter search for $\epsilon$? Additionally, when training data is available, although a single pass of EvA is computationally inexpensive, the hyperparameter sweep over $\epsilon$ may be time consuming. Discussing these, especially, the former case of selecting $\epsilon$ in the absence of an unbiased train subset is necessary.

[a] Elhage, et al., "Toy Models of Superposition", Transformer Circuits Thread, 2022. \
[b] Radhakrishnan et al., "Mechanism for feature learning in neural networks and backpropagation-free machine learning models", Science 2024.

**Questions:**

Please see the Weaknesses section.

---

> ### Author Response · Authors · 2024-11-25
>
> > W1. ... the feature erasure in eq. 9 should technically not be achieving what it claims to achieve. By turning full dimensions on/off, it could be - (i) partially turning off some core features that could be in superposition with a spurious feature; and (ii) not fully turning off spurious features, since it may be in superposition with some core feature dimension which is not turned off.
>
> Thank you for your insightful observation regarding the features. Our method detects spuriousness at the channel level, where each “deep feature” corresponds to a channel in the embedding space, as detailed in lines 147–154. In Appendix A.1, we clarify the distinction between spurious/core deep features (features learned by the neural network and represented as individual channels in the embedding space) and spurious/core components (the actual parts of the input data that are inherently spuriously correlated with the target within the dataset). Notably, spurious/core components are intrinsic to the input itself, while spurious/core deep features represent the neural network's learned representations of those components.
>
> We acknowledge that a spurious feature can indeed be a superposition of both spurious and core components. Actually, our Theorems 1 and 2 are developed under this premise and demonstrate that our indices prioritize the erasure of more spurious components when removing a spurious feature, albeit with trade-offs, such as inadvertently erasing some core components. While directly targeting the full representation could improve precision, it would also significantly increase computational complexity. We greatly appreciate your suggestion and recognize it as a promising direction for future work.
>
> > W2. ...How do the authors account for this imbalance when applying the distance metric $d$?
>
> Thank you for your question. The difference in the size of the unbiased dataset and biased dataset will not affect the computation of consistency theoretically since the Wasserstein distance computes the difference between two distributions. However, if we understand correctly, the concern is that a limited unbiased dataset may not sufficiently represent the true distribution of ideal unbiased data, potentially skewing the consistency measure d. In these cases, evidence energy may be a more reliable proxy for reducing spurious features. For example, as shown in Table 3, with an unbiased dataset of only 16 samples, EvA-E outperforms EvA-C. In practice, however, EvA-C remains effective even with small unbiased datasets. For instance, in the CelebA dataset, where the unbiased dataset comprises only 0.8\% of the biased dataset (as detailed in Table 1), EvA-C still demonstrates strong performance, as seen in Table 5.
>
> > W3. ... However, whether consistency and evidence energy are really the best metric to look at when it comes to performing this selection needs to be either theoretically or empirically established ... erasure mechanisms.
>
> Thank you for the suggestion. We have compared our method with Lasso Regression, as used by DFR (Appendix E.4), and with other potential spurious indicators (Appendix Figure 6). These comparisons highlight the effectiveness of our spurious indicator, demonstrating that its performance is not solely due to sparsity. Additionally, we provide further results of random feature dropout on the BAR dataset using the same protocol as described in the experimental section, achieving an accuracy of only 61.2%.
>
> > W4. Table 5 shows that the gains from EvA are more significant in the absence of extra unbiased training data. The authors should provide a discussion on why their method might have a stronger advantage over SOTA when unbiased training data is not present.
>
> We mainly compare the strength of our methods in the absence of extra unbiased training data in terms of computation efficiency and data efficiency introduced in Section 4.2 and Appendix E.2. In terms of the improvement over accuracy, we have a separated discussion in Appendix. E.1 (Table 6) on how our methods improve over previous baseline in terms of each classes.
>
> > W5. When extra training data is not available, how does one perform the hyperparameter search for ϵ? Additionally, when training data is available, although a single pass of EvA is computationally inexpensive, the hyperparameter sweep over ϵ may be time-consuming. Discussing these, especially, the former case of selecting ϵ in the absence of an unbiased train subset is necessary.
>
> When an extra unbiased dataset is unavailable, we use the biased validation dataset for hyperparameter search, focusing on mean accuracy rather than worst accuracy (as shown in Table 4). In practice, since embeddings are pre-extracted for linear layer fitting, scanning all possible erase ratios is computationally efficient. As noted in lines 411–412, the total runtime for Waterbirds, including searching on 90 candidates of hyperparameter, is only 10 minutes. We will clarify this in the revision.

---

> > ### Comment · Reviewer_Jy78 · 2024-12-03
> > **All concerns have been addressed by the authors**
> >
> > I thank the authors for their work towards the rebuttal. All the concerns that I had raised in my initial reviews have been adequately addressed. I have thus increased my score.

---

### Meta-Review · Area_Chair_eeTu · 2024-12-20

**Metareview:**

The paper proposes EwA, a feature selection method for removing reliance on spurious correlations. The method builds on the observations from [1] that information about both core and spurious features is contained in the penultimate layer representations. The authors propose a simple rule for deciding on which features to drop based on (1) difference in feature distributions between train and unbiased validation (when available) or (2) influence on model confidence. The authors show theoretically in a simple setting that these choices are justified. Then, they show that the proposed method performs strongly on benchmarks, especially when the unbiased rebalancing dataset is small or unavailable.

Strengths:
- The method is very simple and makes intuitive sense.
- The method is computationally very cheap.
- The method shows especially strong performance in the setting where the rebalancing data is extremely limited.
- The authors provide theoretical motivation for the proposed method.
- The authors provide detailed ablations on hyper-parameters, as well as understanding experiments showing a correlation between the two feature selection criteria proposed.
- The method performs well across a broad range of tasks.

Weaknesses:
- The method assumes that the "true features" are aligned with dimensions of the penultimate layer representations. This is known to not be true in the context of LLMs [2]. However, this may still be true in image classification models such as ResNet, considered by the authors.
- The feature selection for the setting where rebalancing data is not available relies on the standard assumption that spurious features are the most predictive in the classifier. While true on benchmarks, this may not hold on more realistic problems, where spurious features may be weaker than core features.
- The authors only evaluate the method on image classification tasks. The standard set of evaluations in spurious correlations literature typically also includes CivilComments and MultiNLI NLP datasets. It would be interesting to see results there. Generally, it would be nice to expand the set of evaluation datasets.
- The main results table 5 uses a non-standard setting with ResNet-18 instead of ResNet-50. The authors mention that on ResNet-50, with the full reweighting dataset, the method does not outperform DFR [1], but in Table 5 it does. It may be better to report the results in the standard evaluation setting in the main text.
- Reviewers highlighted some related work that should be included in the paper.

Decision recommendation: The paper proposes a very simple and intuitive method that works well in practically relevant settings. The strong performance when the rebalancing dataset is small is very promising. I believe this paper makes a good contribution and recommend to accept it.


[1] Last Layer Re-Training is Sufficient for Robustness to Spurious Correlations
Polina Kirichenko, Pavel Izmailov, Andrew Gordon Wilson

[2] [Toy Models of Superposition](https://transformer-circuits.pub/2022/toy_model/index.html)

**Additional Comments On Reviewer Discussion:**

The reviewers unanimously vote to accept the paper: 8, 6, 6, 6. They raised concerns about the details of the evaluation, and the authors provided additional results and experiments. As a result, two of the reviewers raised their scores, and the other reviewers responded that they were satisfied with the rebuttal.

---

### Decision · Program_Chairs · 2025-01-22

Accept (Poster)